# Oviductin sets the species-specificity of the mammalian zona pellucida

Daniel de la Fuente[1,2†], Maria Maroto[1†], Yulia N Cajas[1,3,4], Karina Canon-Beltran[1,5], Raul Fernandez-Gonzalez[1], Ana Munoz-Maceda[6], Juana M Sanchez-Puig[7], Rafael Blasco[7], Paula Cots-Rodríguez[8], Manuel Avilés[8], Dimitrios Rizos[1], Alfonso Gutierrez-Adan[1]*

[1]Department of Animal Reproduction, INIA-CSIC, Madrid, Spain; [2]FivCenter Madrid Clinics, Aravaca,, Madrid, Spain; [3]Dept. Agrarian Production, Technical University of Madrid, Madrid, Spain; [4]Laboratorio de Biotecnología de la Reproducción Animal, Facultad de Ciencias Agropecuarias, Universidad de Cuenca (UC), Cuenca, Ecuador; [5]Programa de Medicina Veterinaria y Zootecnia, Grupo Kyron, Corporación Universitaria del Huila (CORHUILA), Huila, Colombia; [6]Department of Animal Medicine and Surgery, Veterinary Faculty, UCM, Madrid, Spain; [7]Departamento de Biotecnología, INIA-CSIC, Madrid, Spain; [8]Departamento de Biología Celular e Histología, Universidad de Murcia-Instituto Murciano de Investigación Biosanitaria Pascual Parrilla, Murcia, Spain

*For correspondence: agutierr@inia.csic.es

†These authors contributed equally to this work

Competing interest: The authors declare that no competing interests exist.

## eLife Assessment

This **valuable** study unravels the mechanisms underlying mammalian sperm-oocyte recognition and penetration, shedding light on cross-species interactions. It provides **solid** evidence that exposure of sperm to oviductal fluid or OVGP1 proteins from bovine, murine, or human sources imparts species-specific zona pellucida (ZP) recognition, ensuring that only sperm from the corresponding species can penetrate the ZP, regardless of its origin. These findings hold significant potential for reproductive biology, offering insights to enhance porcine in vitro fertilization (IVF), which frequently suffers from polyspermy, as well as advancing human IVF through improved intrinsic sperm selection.

**Abstract** The zona pellucida (ZP) is vital for species-specific fertilization as this barrier mediates sperm-oocyte binding. Here, we determined whether sperm from distant mammalian orders (Carnivora, Primates, and Rodentia) could penetrate bovine oocytes by examining the role of bovine oviductal fluid and species-specific oviductal glycoprotein (OVGP1 or oviductin) from bovine, murine, or human sources in modulating the species-specificity of bovine and murine oocytes. Sperm from all the species were found to penetrate intact bovine ovarian oocytes to form hybrid embryos. However, contact with oviductal fluid or bovine, murine, or human OVGP1, conferred the ZP species-specificity, allowing only the penetration of the corresponding sperm regardless of the ZP's origin. Glycolytic and microstructural analyses revealed that OVGP1 covers the pores present in the ZP and that OVGP1 glycosylation determines sperm specificity. This suggests specific fertilization capacity is acquired in the oviduct through the ZP's incorporation of specific oviductin.

## Introduction

Interaction between the sperm and egg is generally assumed to be a species-specific event mediated by the recognition and binding of complementary molecules on the sperm plasma membrane

to both the zona pellucida (ZP) and the oocyte oolemma (*Vacquier, 1998*; *Vieira and Miller, 2006*). However, it remains unclear whether both the oocyte's ZP and plasma membrane are necessary to determine species-specificity. In evolutionary terms, the ZP is the oldest among the coatings that envelop vertebrate oocytes and conceptuses. This barrier is responsible for preventing polyspermy, facilitating preimplantation development, and mediating species-restricted recognition between the gametes (*Moros-Nicolás et al., 2021*). ZP proteins and sperm-oocyte binding protein are usually found to evolve under selection pressure. It is proposed that this rapid evolution plays an important role in the reproductive isolation of diverging taxa (*Meslin et al., 2012*; *Pisciottano et al., 2024*). The mechanism of fusion between the sperm and oolemma membranes is not fully understood, although various candidate molecules have been identified (*Inoue et al., 2005*; *Lamas-Toranzo et al., 2020*; *Noda et al., 2020*). The species-specificity of interactions between gamete membranes in mammals is not always apparent. The significance of the ZP in maintaining species boundaries is revealed by the fact that its removal allows for heterospecific sperm binding and penetration of the oocyte plasma membrane. For example, sperm from humans, mice, rats, guinea pigs, goats, bulls, horses, pigs, rabbits, and dolphins can all successfully penetrate hamster oocytes lacking their ZP coat (*Vieira and Miller, 2006*; *Yanagimachi and Phillips, 1984*). This phenomenon was demonstrated in the hamster egg penetration test (HEPT) developed by *Yanagimachi et al., 1976*. In this test, researchers observed that upon removal of the ZP from hamster ova, the eggs allowed for the penetration of sperm from other species (*Yanagimachi et al., 1976*). However, it remains unknown why zona—free hamster eggs are readily penetrated by heterologous spermatozoa, as this characteristic is not shared by the ova of other species. For instance, human spermatozoa will not penetrate zona—free rat, rabbit, or mouse ova (*Prasad, 1984*). The ability of sperm from different species to penetrate the hamster oocyte suggests that the ZP could be a key factor in preventing heterologous fertilization between different species.

The successful in vitro fertilization (IVF) of zona—free bovine oocytes with deer spermatozoa has been reported (*Comizzoli et al., 2001*). The IVF of zona-intact bovine oocytes with endangered African antelope spermatozoa was also described in 1998 (*Roth et al., 1998*). Further heterologous gamete interactions have been documented between porcine or equine sperm and ZP-intact bovine oocytes in large domestic animals (*Sinowatz et al., 2003*), as have interspecific interactions between equine and porcine gametes (*Mugnier et al., 2009*). However, other research groups have reported complete species-specificity for IVF between heterologous gametes in pigs and cattle (*Takahashi et al., 2013*). In prior work, we observed in zona-intact bovine oocytes obtained directly from ovaries, the sperm fertility of other species of the same order, such as ram (*García-Alvarez et al., 2009*), common bottle-nose dolphins (*Sánchez-Calabuig et al., 2015*), and wild goats (*Pradieé et al., 2018*). This suggests that species within the order Artiodactyla share similar fertilization mechanisms. However, it appears that this capacity is not restricted solely to this order. Both our group and other researchers have found that horse (order Perissodactyla) and guinea pig spermatozoa (order Rodentia) are also capable of fertilizing zona-intact bovine oocytes (*Sessions-Bresnahan et al., 2014*; *Consuegra et al., 2020*; *Cañón-Beltrán et al., 2023*). In the present study, we used spermatozoa from the orders Carnivora (cats), Primates (humans), and Rodentia (mice, Muridae family) to examine whether species from other mammalian orders can also fertilize zona-intact bovine oocytes.

The ZP is an extracellular matrix that surrounds the oocyte and preimplantation embryo in mammals (*Florman and Storey, 1982*; *Litscher and Wassarman, 2020*). This matrix acts as a barrier, separating cumulus cells from the oocyte, facilitating species-restricted recognition during fertilization, and preventing polyspermy, contributing in this way to preimplantation embryo development (*Moros-Nicolás et al., 2021*). ZP matrix proteins are glycosylated and specific oligosaccharide residues have been linked to their role as sperm receptors (*Benoff, 1997*). The diverse glycoconjugate composition of the ZP is thought to be unique to each species (*Moros-Nicolás et al., 2021*; *Avilés et al., 1994*). In cattle, the main components (77%) of the carbohydrate chains of ZP glycoproteins are acidic chains containing sialic acids (*Katsumata et al., 1996*): mainly Neu5Ac (84.5%) and Neu5Gc (15.5%) linked to Galβ1-4GlcNAc and GalNAc by α2,3- and α2,6-linkages on N- and O-glycans respectively (*Velásquez et al., 2007*). Oviductal fluid composition plays a crucial role in fertilization (*Avilés et al., 2010*) and subsequent embryo development (*Maillo et al., 2015*; *Lopera-Vasquez et al., 2017*; *Hamdi et al., 2018*). Oviductal proteins have been proposed to induce changes in the ZP, influencing sperm-oocyte binding specificity (*Pérez-Cerezales et al., 2018*; *Maillo et al., 2016*). These changes may impact

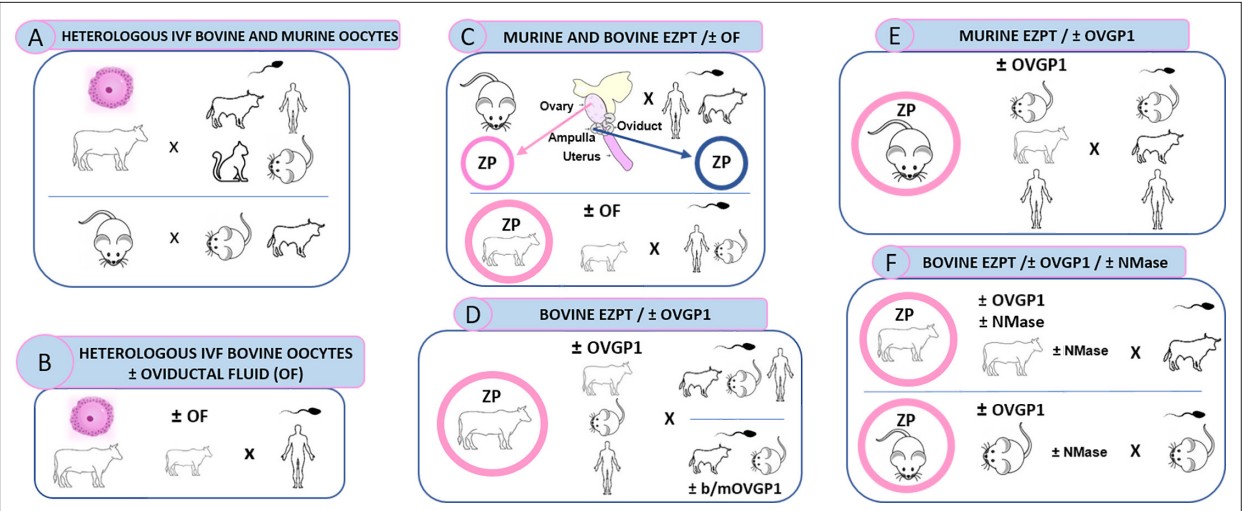

**Figure 1.** Schematic representation of global experimental design of homologous and heterologous IVF from diverse mammalian species. (**A**) Represents design of experiment shown in *Appendix 1—table 1*, *Appendix 1—table 2*, *Appendix 1—table 3*, *Figure 2A and B*, *Figure 2—figure supplement 1*, *Figure 2—figure supplement 2* and *Figure 2—figure supplement 3*. (**B**) Represents design of experiment shown in *Appendix 1—table 4*, *Figure 2C*. (**C**) Represents design of experiment shown in *Appendix 1—table 5*, *Appendix 1—table 6*, *Figure 2D and E* and *Figure 3*. (**D**) Represents design of experiment shown in *Appendix 1—table 8*, *Appendix 1—table 9*, *Appendix 1—table 10*, *Figure 5* and *Figure 7A*. (**E**) Represents design of experiment shown in *Appendix 1—table 11*, and *Figure 7B*. (**F**) Represents design of experiment shown in *Appendix 1—table 12*, *Appendix 1—table 13*, *Appendix 1—table 14* and *Figures 8 and 9*.

ZP maturation, sperm-ZP binding, and the prevention of polyspermy (*Avilés et al., 1994*). Modifications induced by oviductal proteins could alter the ZP's composition by affecting its carbohydrates or proteins. One set of proteins found in oviductal fluid is a family of glycosylated proteins collectively known as oviduct-specific glycoprotein (OVGP1, also known as oviductin). This protein has been identified in various mammalian species, including humans. Oviductin has been associated with the ZP of ovarian oocytes post-ovulation (*Zhao et al., 2022*). OVGP1, along with heparin-like glycosaminoglycans (GAGs), plays a role in functionally modifying the ZP in pigs and cows, making it more resilient to enzyme digestion and sperm penetration, thereby helping prevent polyspermy (*Coy et al., 2008*; *Algarra et al., 2018*). However, it remains uncertain whether the ZP's sperm receptor is consistent across all mammalian species. This raises questions about how sperm from different mammalian orders are able to fertilize cow oocytes, and whether the lack of contact between cow oocytes from slaughterhouse ovaries and oviductal proteins prevents the changes in the ZP necessary for bull sperm recognition. Here, we have explored the role of bovine oviductal fluid and murine, human, or bovine oviductal glycoprotein 1 (OVGP1) in modulating the sperm-binding species-specificity of the bovine and murine ZP.

## Results

### Human, mouse, and cat spermatozoa can fertilize bovine ovarian oocytes

A schematic depiction of the experimental design is indicated in *Figure 1*.

We first examined whether human spermatozoa could penetrate bovine oocytes obtained directly from the ovary, without prior exposure to oviductal fluids. For this purpose, human spermatozoa were capacitated in two different media: G-IVF PLUS (HeA) and Fert (HeB), commonly used for the capacitation of human spermatozoa and bovine spermatozoa, respectively. As presented in *Appendix 1—table 1* and depicted in *Figure 2*, *Figure 2—figure supplement 1*, human spermatozoa were able to penetrate the bovine oocytes, which went on to develop a pronucleus and eventually underwent cleavage into the two-cell stage. Heterologous fertilization outcomes were improved when the G-IVF PLUS medium (HeA) was used instead of Fert (HeB). However, these outcomes were still worse than those obtained for homologous fertilization. No differences were observed in the number of

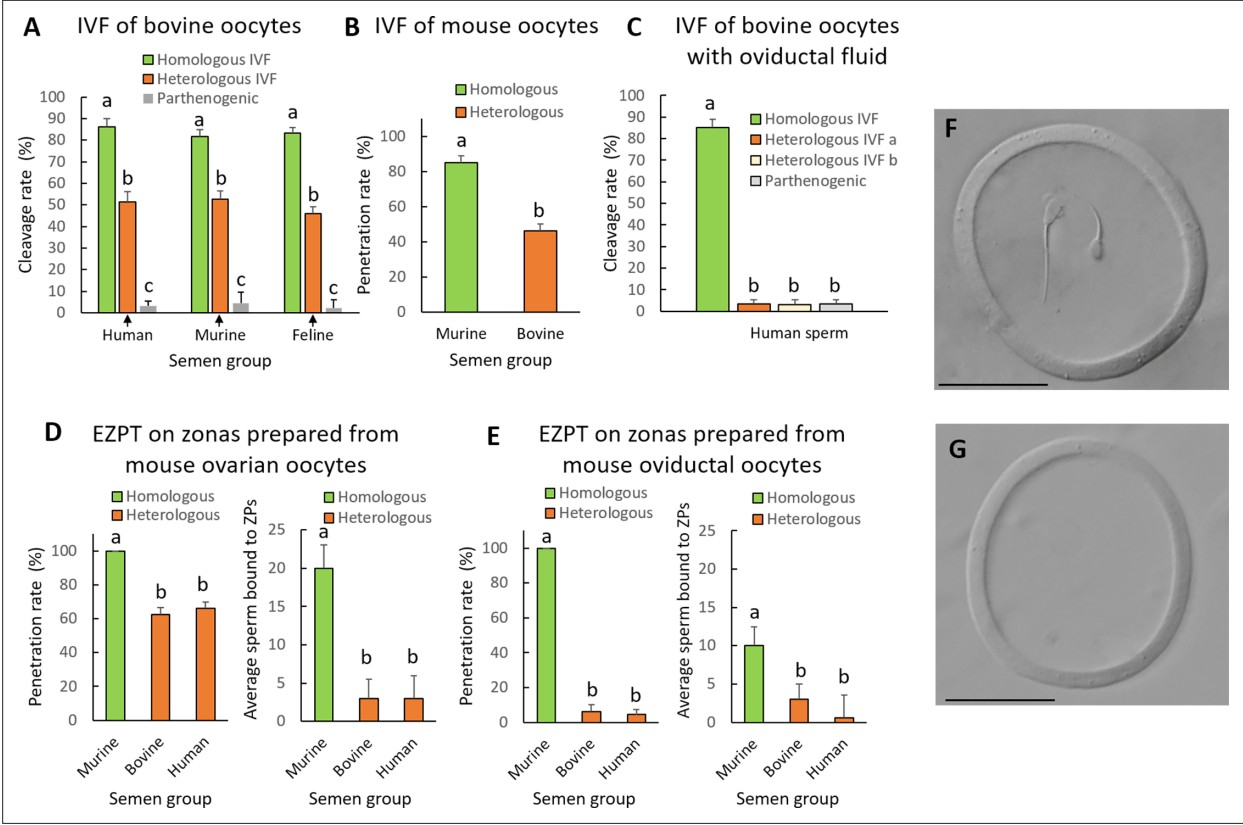

**Figure 2.** Heterologous IVF of bovine oocytes, mouse oocytes, or empty mouse ZPs, using human, mouse, or cat sperm, before and after contact with oviductal fluid. (**A**) Embryo cleavage rates resulting from the IVF of bovine oocytes with human, murine, or feline sperm including bovine sperm as homologous IVF control, and parthenogenesis as a negative control of the IVF. (**B**) Sperm penetration rate after the IVF of IVM mouse oocytes with murine or bovine sperm, for homologous and heterologous IVF, respectively. (**C**) Embryo cleavage rates resulting from the IVF of bovine IVM oocytes, preincubated 30 min with bovine oviductal fluid, with human sperm IVF medium a=G-IVF PLUS medium (HeA); IVF medium b=Fert (HeB). (**D**) Penetration rates and average numbers of sperm bound to ZP after the empty zona penetration test (EZPT) using mouse ovarian IVM oocytes and murine, bovine, or human sperm. (**E**) Penetration rates and average numbers of sperm bound to ZP after the EZPT using mouse oviductal oocytes and murine, bovine, or human sperm. (**F**) Picture of an empty zona pellucida obtained from a mouse ovarian IVM oocyte after the EZPT using bovine sperm. Note the sperm has penetrated the zona. (**G**) Picture of an empty zona pellucida from a mouse oviductal oocyte after the EZPT using bovine sperm. Scale bar for ZP pictures = 50 µm. A non—fertilized parthenogenesis group is used as cleavage control in (**A**) and (**C**). Different letters above error bars (mean ± SD) indicate significant differences (p<0.05) among groups (ANOVA and Tukey's post hoc test). Numbers of oocytes or ZPs used are indicated in *Appendix 1—table 1*, *Appendix 1—table 2*, *Appendix 1—table 3*, *Appendix 1—table 4* and *Appendix 1—table 5*.

The online version of this article includes the following figure supplement(s) for figure 2:

**Figure supplement 1.** Heterologous IVF between bovine oocytes and human sperm.

**Figure supplement 2.** Heterologous IVF between bovine oocytes and murine sperm.

**Figure supplement 3.** Heterologous IVF between bovine oocytes and cat sperm.

**Figure supplement 4.** Steps (**A–E**) of the method used to empty the bovine zone pellucida by removing the cytoplasmic contents of the oocyte containing all the organelles, nucleus and membranes, and removing also the polar body.

spermatozoa remaining attached to the ZP, or in pronuclear formation up to 12 hours post-incubation (hpi). These results indicate that human spermatozoa can bind to and penetrate both the bovine oocyte's ZP and plasma membrane. Human spermatozoa were also found capable of inducing the first embryo cleavage.

As shown in *Appendix 1—table 2* and illustrated in *Figure 2*, *Figure 2—figure supplement 1*, mouse spermatozoa showed a similar capacity to human spermatozoa to fertilize bovine ovarian oocytes. Following sperm penetration of the oocyte, the sperm nucleus decondensed, forming a pronucleus, which ultimately induced cleavage into two-cell stage bovine zygote. Pronucleus formation and cleavage rates were, however, lower than those obtained for homologous fertilization.

Nonetheless, these findings confirm that mouse spermatozoa can also bind to and penetrate both the bovine oocyte ZP and plasma membrane, and are also capable of inducing the first embryo cleavage.

Similar results were also obtained with cat sperm. As presented in *Appendix 1—table 3* and depicted in *Figure 2*, *Figure 2—figure supplement 3*, feline spermatozoa were found to bind to and penetrate bovine oocytes, thereby initiating the first cellular division of the hybrid embryo. Pronucleus formation and cleavage rates were lower than those observed for homologous bovine fertilization.

To explore if the same effect was produced using oocytes from another species, we collected oocytes from the ovaries of mice following PMSG-induced stimulation. The ovarian murine oocytes were matured in vitro and then fertilized with bovine sperm. The results obtained were similar to those obtained with bovine oocytes in that both the ZP and plasma membrane of the mouse oocytes could be penetrated by the bovine sperm, which then underwent decondensation giving rise to two-cell cleavage stage embryos (*Figure 2B*).

## Oviductal fluid inhibits the heterologous fertilization of bovine ovarian oocytes

We then went on to investigate whether the interaction of oocytes with oviductal fluid would hinder the heterologous fertilization of bovine oocytes with human spermatozoa. In this experiment, oocytes were subjected to 30 min of incubation with oviductal fluid derived from the oviducts of slaughtered heifers. Subsequently, these oocytes were incubated with either human or bovine spermatozoa. Results showed that oocytes co-incubated with oviductal fluid could only be fertilized by bovine spermatozoa, with 86% of these oocytes cleaving into two cells. In contrast, bovine oocytes that had been exposed to oviductal fluid were impervious to the penetration by human spermatozoa when using either of the two IVF media (only 3% of these oocytes cleaved into two cells, a similar rate to that recorded in the non—fertilized parthenogenesis group; see *Appendix 1—table 4* and *Figure 2C*). These findings suggest that the ZP and/or oolema undergo certain alterations upon contact with oviductal fluid, rendering them penetrable exclusively by spermatozoa of the same species. No sperm were found in the perivitelline space, suggesting that the changes occur in the ZP.

## Oviductal fluid sets the species-specificity of the zona pellucida

To validate our hypothesis that the ZP undergoes modifications within the oviduct that confer species-specificity, we used murine oocytes sourced both from the ovary and oviduct. We extracted all the contents inside of the ZP of the oocyte (the cytoplasmic containing all the organelles, nucleus and membranes, and the polar body; *Figure 2—figure supplement 4*), and then subjected them to either homologous IVF with mouse sperm or heterologous IVF with bovine or human sperm over 4 hr. Because this experiment was done on empty ZPs, we called this test 'empty zona penetration test' (EZPT). Our results showed that the ZPs of oocytes obtained directly from the ovary were susceptible to penetration by the sperm of all three species, but once exposed to oviductal fluids, they could only be penetrated by sperm of their homologous species (see *Appendix 1—table 5* and *Figure 2D–G*). While mouse spermatozoa were found to adhere to the ZP of oocytes obtained both from the ovary and oviduct, more sperm cells appeared on the ZP of oocytes from the ovary. Further, very few bovine and human spermatozoa adhered to the ZP of mouse oocytes obtained from the ovary or oviduct (*Appendix 1—table 5* and *Figure 2D and E*).

To test whether oviductal fluid indeed determines the species-specificity of bovine oocytes and ascertain whether bovine oviductal fluid affects the ZP, the oocyte, or both, we extracted the oocyte contents, leaving only the ZP (*Figure 2—figure supplement 4*) and exposed the empty bovine ZP to oviductal fluid for 30 min. Subsequently, we co-incubated ZP with either bovine, human, or murine spermatozoa for a period of 4 hr in an EZPT. As observed in *Figure 3*, when ZPs were obtained from ovarian oocytes and not exposed to the oviductal fluid, although the numbers of spermatozoa penetrating the ZP were higher in the homologous IVF group, penetration percentages were similar for human, murine or bovine spermatozoa (see *Appendix 1—table 6* and *Figure 3A, C, E and G*). However, when ZP devoid of oocyte cytoplasmic contents were exposed to oviductal fluid for 30 min, these were only susceptible to penetration by bovine spermatozoa (see *Appendix 1—table 6* and *Figure 3B, D, F and H*), thereby confirming that the effect of oviductal fluid on the ZP is sufficient to determine the specificity of penetrating sperm.

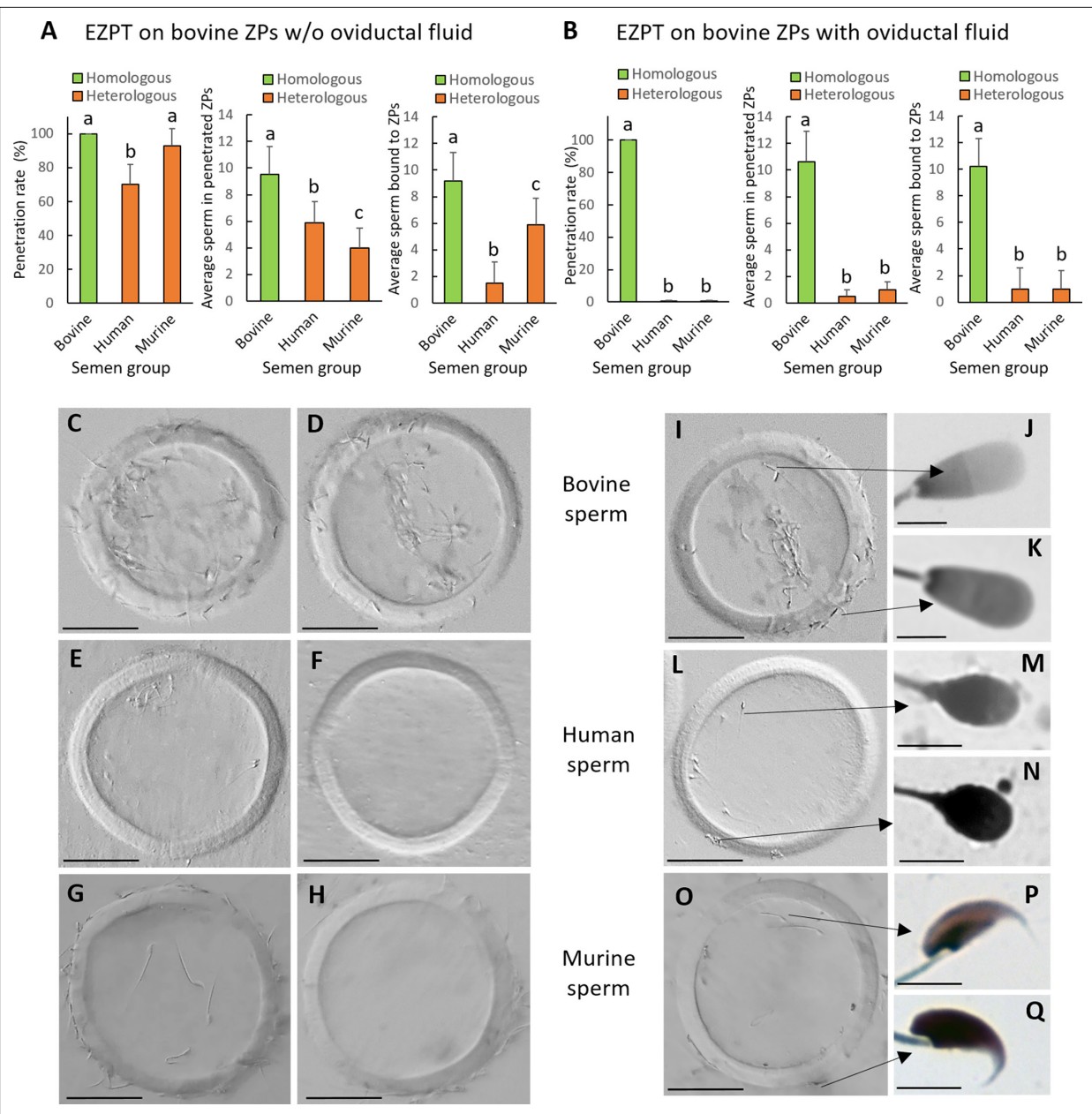

**Figure 3.** Incubation of empty ZPs obtained from bovine ovarian oocytes with oviductal fluid determines the specificity of spermatozoa capable of penetrating the zona. The empty zona penetration test (EZPT) in ZPs obtained from IVM bovine oocytes was performed after homologous (bovine sperm) or heterologous (human or murine sperm) fertilization. Similar outcomes were observed when ZPs were not treated with oviductal fluid (**A**), but after incubation with oviductal fluid for 30 min, human and murine sperm were unable to penetrate the bovine ZPs (**B**) (six replicates per medium per semen sample). Different letters above error bars (mean ± SD) indicate significant differences ($P<0.05$) among groups (ANOVA and Tukey's post hoc test). Numbers of ZPs used are indicated in *Appendix 1—table 6*. (**C, D, I**) Representative pictures of empty bovine ZPs penetrated by bovine sperm illustrating that the penetrating sperm (**J**) lack an acrosome, while those unable to penetrate the zona maintain the acrosome (**K**). (**E, F, L**) Empty bovine ZP penetrated by human sperm revealing that the penetrating sperm have lost the acrosome (**M**), whereas those not penetrating the zona maintain the acrosome (**N**). (**G, H, O**) Representative pictures of empty bovine ZPs penetrated by murine sperm illustrating that the penetrating sperm (**P**) lack an acrosome, while those unable to penetrate the zona maintain the acrosome (**Q**). In the absence of oviductal fluid, the empty bovine ZP can be penetrated by bovine (**C, I**), human (**E, L**), or murine (**G, O**) sperm; however, when the ZP has been in contact with oviductal fluid, it can only be penetrated by bovine sperm (**D**), and not by human (**F**) or mouse (**H**) sperm. Scale bar for ZP pictures = 50 µm. Scale bar for sperm pictures = 5 µm.

With our EZPT, we were able to determine that all spermatozoa penetrating the ZP lacked an acrosome, while the majority of those that failed to penetrate the egg coat retained an intact acrosome (see *Figure 3I–Q*). To assess acrosomal status, the fluorescein isothiocyanate-conjugated peanut agglutinin (FITC-PNA) method was used (*Fernandez-Fuertes et al., 2017*). This observation suggests that during the penetration of the ZP, bovine, murine, and human spermatozoa undergo the acrosomal reaction. Additionally, it should be noted that, regardless of incubation with oviductal fluid, a higher proportion of bovine spermatozoa remained bound to the ZP (see *Appendix 1—table 6* and *Figure 3C, D, I*). It may be seen in *Appendix 1—table 6* and *Figure 3A and B*, that more bovine spermatozoa bound to the ZP than human and murine ones. No difference was found between numbers of bovine sperm bound to ZP treated or not with oviductal fluid. Also, for human spermatozoa, no difference was found between those bound to ZPs treated or not with oviductal fluid, but for mice sperm, a greater number attached to the bovine ZP when it had not been in contact with oviductal fluid.

## Homology of bovine, human, and murine OVGP1 proteins and binding of recombinant OVGP1 proteins with murine or bovine ZPs

It has been reported that OVGP1 in pigs influences sperm binding to the ZP in a species-specific manner (*McCauley et al., 2003*) and that this protein facilitates specific sperm binding to the ZP, potentially contributing to species-specificity (*Avilés et al., 2010*; *Pérez-Cerezales et al., 2018*). To investigate the role of OVGP1 on the species-specificity of the ZP, we have use histidine—Tagged cow recombinant OVGP1 glycoproteins and DDK—Tagged human and mouse recombinant OVGP1 glycoprotein. As depicted in *Figure 4A*, the sequences of these three OVGP1 have five distinct regions (A, B, C, D, and E; *Avilés et al., 2010*). Regions A and C are conserved in the different mammals. Region A, corresponding to the amino terminal end, shows high identity among monotremes, marsupials, and placentals. Region B features low identity among the different mammals and contains multiple insertions/deletions. Region D is typical of human oviductin and region E is an insertion present only in the mouse (Mus). Details of OVGP1 sequence alignments for the three species can be seen in *Figure 4—figure supplement 1*; *Zhao et al., 2022*; *Choudhary et al., 2019*; *Buhi, 2002*. Clear differences may be seen in predicted N-glycosylation sites and chitin-binding sites (*Figure 4—figure supplement 1*) as well as in predicted O-glycosylation sites (*Appendix 1—table 7*).

Proteins were expressed in mammalian cells, separated by SDS-PAGE and analyzed by immunoblotting using rabbit polyclonal antibody to human OVGP1. Additionally, rabbit monoclonal antibody against His—Tag, and mouse monoclonal antibody against Flag (DDK)-Tag immunoblots were performed to test the recombinant OVGP1 proteins (data not shown). Western blots of the three OVGP1 recombinants indicated expected sizes based on those of the proteins: 75 kDa for human and murine OVGP1 and around 60 kDa for bovine OVGP1 (*Figure 4B* and *Figure 4—figure supplement 2*). Immunoblots were prepared of bovine and murine oviductal fluid protein extracts from animals in oestrus (mice) or ovulated (cow) versus anoestrus animals. OVGP1 in oviductal fluids was in the range of 100–150 ng/μL in mice and 150–200 ng/μL in cow. In *Figure 4B*, an arrow indicates the presence of the expected band of OVGP1 found in the oviductal fluid of female mice in oestrus or oviduct of ovulated cows, compared to the oviductal fluid of animals in anoestrus. Murine oviductal fluids displayed the expected 75 kDa band and a more intense 50 kDa band, but only the 75 kDa band corresponded to OVGP1 glycoprotein, as confirmed by proteomics of the bands along with PEAKS Studio v11.5 search engine peptide identification software. As according to Proteomic results, in the 75 kDa band OVGP1 was identified with 14 unique peptides over 240 protein groups, whereas in the 50 kDa band it was not present OVGP1 over 307 identified protein groups.

OVGP1 recombinant proteins were also assessed for their ability to bind murine or bovine ZPs obtained from in vitro matured (IVM) ovarian oocytes compared to ZPs derived from murine oocytes from superovulated females (*Figure 4*, *Figure 4—figure supplement 3*). Oocytes were incubated for 30 min at room temperature (RT) with bovine (bOVGP1), murine (mOVGP1), or human (hOVGP1) OVGP1, and were then fixed and imaged by confocal fluorescence and DIC microscopy using rabbit polyclonal antibody to human OVGP1 for bOVGP1 and endogenous OVGP1, and mouse monoclonal antibody against Flag (DDK)—Tag for hOVGP1 and mOVGP1. All three recombinant proteins were able to bind to the murine and bovine ZPs (*Figure 4C and D*; controls in *Figure 4—figure supplement 3*). No differences were observed in the ability of the three OVGP1s to attach and penetrate the mouse ZPs (*Figure 4C*, *Figure 4—figure supplement 3A*). However, while binding of bOVGP1 was

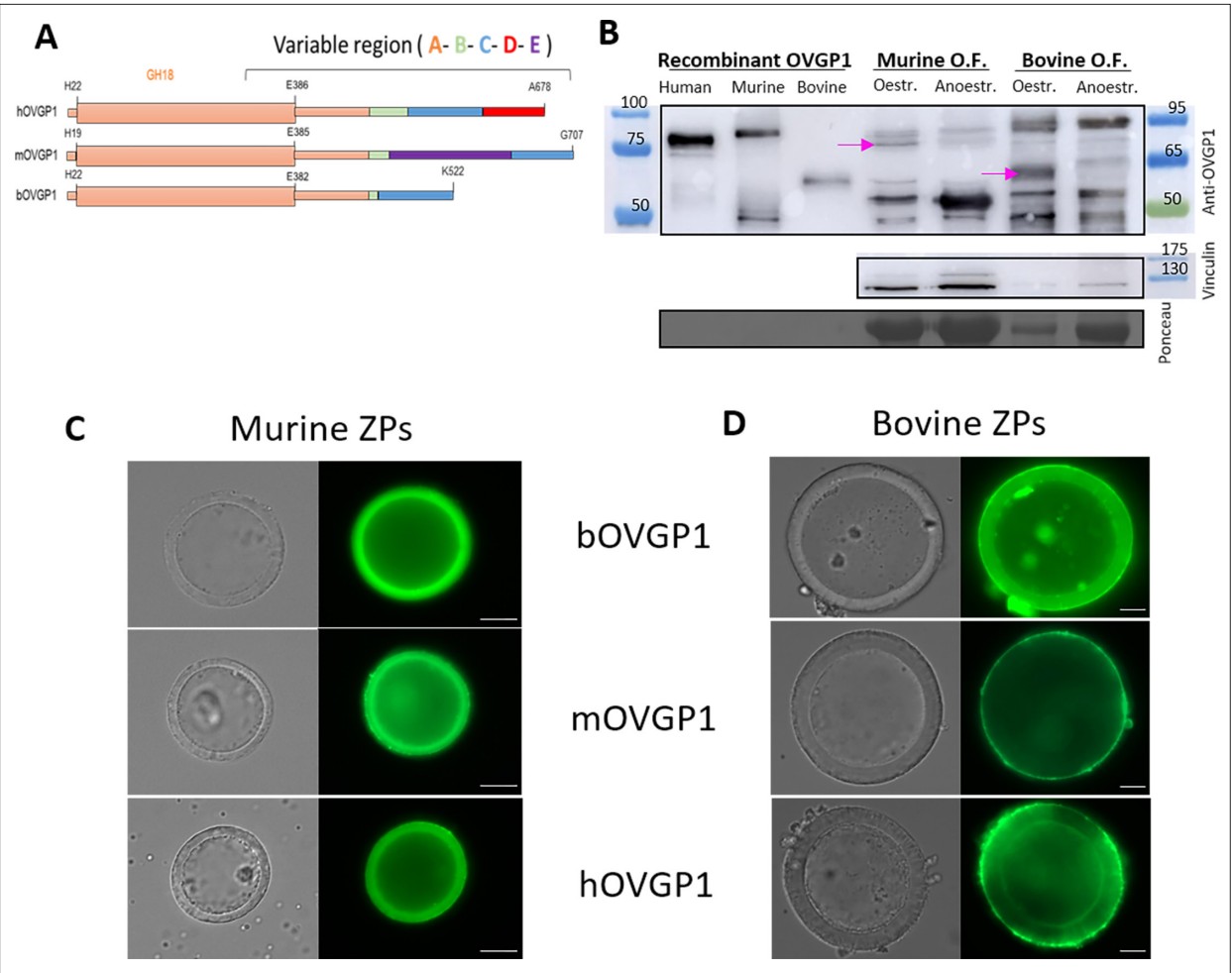

**Figure 4.** Structure of oviductin, western blots of OVGP1 recombinants and localization of these recombinants at the bovine or murine ZPs. (**A**) Diagram showing the five regions (A, B, C, D, and E) present in some of the oviductin proteins of human (hOVGP1), murine (mOVGP1), and bovine (bOVGP1) mammalian species (adapted from Figure 1 from *Avilés et al., 2010*). (**B**) Western blots of the three OVGP1 recombinants proteins used in this study (human, murine, and bovine). Recombinant bovine protein was expressed in BHK-21 cells and purified, whereas recombinant murine and human proteins were purchased by Origene and had been produced in HEK293T. Then, proteins were separated by SDS-PAGE and analyzed by immunoblotting using rabbit polyclonal antibody to the human OVGP1. The following lanes of the gel contain the protein extracts from oviductal fluid of female mice in oestrus and anoestrus and from oviductal fluid of ovulated cows or anoestrus cows, indicating with an arrow the presence in estrous of the OVGP1 band for both species. ZPs from IVM murine (**C**) and bovine (**D**) oocytes were incubated for 30 min at RT with recombinants bOVGP1, mOVGP1, and hOVGP1. ZPs were fixed and imaged by confocal fluorescence and DIC microscopy using rabbit polyclonal antibody to the human OVGP1 for bOVGP1, and a monoclonal antibody against Flag—Tag for hOVGP1 and mOVGP1. Scale bars = 20 µm.

The online version of this article includes the following source data and figure supplement(s) for figure 4:

**Source data 1.** Labeled data from *Figure 4B*.

**Source data 2.** Raw data from *Figure 4B*.

**Figure supplement 1.** Sequence alignments of OVGP1 from *Homo sapiens* (NCBI AAI36407.1), *Bos taurus* (NP_001073685.1), and *Mus musculus* (AAI37996.1).

**Figure supplement 2.** Recombinant OVGP1 from human, murine and bovine recognized with anti-OVGP1 (**A**) and anti-His/anti-Flag (**B**).

**Figure supplement 2—source data 1.** Labeled data from *Figure 4—figure supplement 2*.

**Figure supplement 2—source data 2.** Raw data from *Figure 4—figure supplement 2*.

**Figure supplement 3.** Representation of immunofluorescence of control ZPs of IVM murine and bovine oocytes, ZPs incubated with murine or bovine oviductal fluid, and ZPs from murine superovulated (SOV).

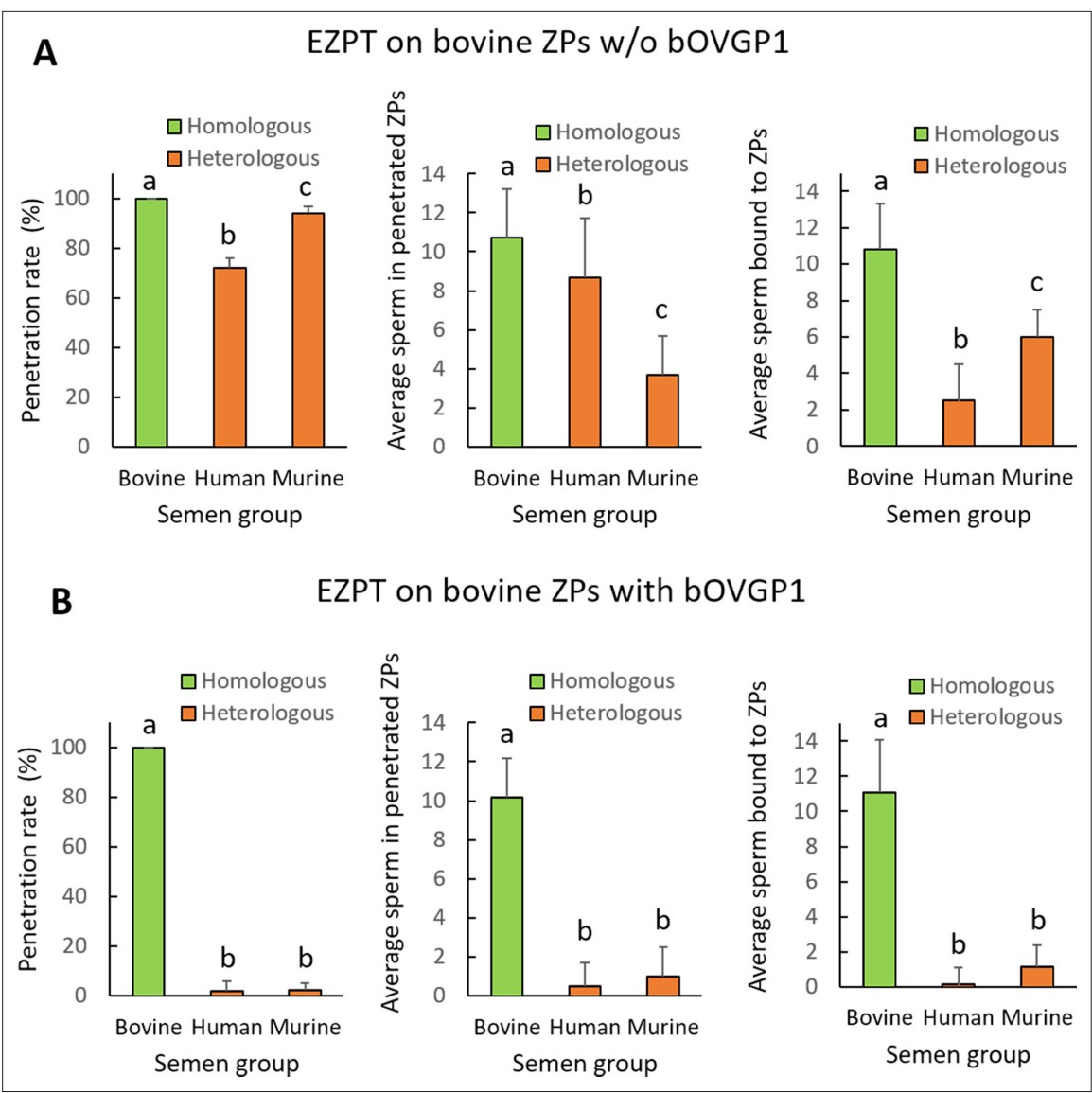

**Figure 5.** Incubation of empty ZPs obtained from bovine ovarian oocytes with OVGP1 determines the specificity of spermatozoa capable of penetrating the zona. The empty zona penetration test (EZPT) was performed after homologous (bovine sperm) or heterologous (human or murine sperm) fertilization. Similar penetration rates were observed when ZPs were not treated with bOVGP1 protein (**A**) but after incubation with bOVGP1 for 30 min, human or murine sperm were unable to penetrate the bovine ZPs (**B**) A drastic reduction was also observed for fertilization by heterologous sperm, both in the average number of sperm penetrating the ZPs, and in the average of number of sperm binding to the ZPs, when fertilization without pretreatment of the ZP with bOVGP1 (**A**) was compared to fertilization after the zona had been in contact with bOVGP1 (**B**). (6 replicates per medium per semen sample). Different letters above error bars (mean ± SD) indicate significant differences (p<0.05) among groups (ANOVA and Tukey's post hoc test). Numbers of ZPs used are indicated in *Appendix 1—table 8*.

observed throughout the thickness of the bovine ZP, mOVGP1 and hOVGP1 preferentially bound to the outer bovine ZP surface without penetrating the coat (*Figure 4D*, *Figure 4—figure supplement 3B*).

## OVGP1 plays a species-specific role in setting the specificity of the zona pellucida

To investigate whether oviductal protein OVGP1 had a similar effect to oviductal fluid in determining the species-specificity of bovine ZP during fertilization, we conducted the EZPT using ZPs from ovarian bovine oocytes that were either treated with bOVGP1 or left untreated (for 30 min). In this experiment, we used homologous (bovine) and heterologous sperm (human and murine). As illustrated in *Figure 5*,

when the ZPs from ovarian oocytes had not been exposed to bovine OVGP1, both homologous and heterologous sperm could penetrate the ZPs (*Appendix 1—table 8* and *Figure 5A*). However, after ZPs were treated with bOVGP1 for 30 min, only bovine sperm were capable of penetrating the ZP, while human or mouse sperm were unable to do this (*Appendix 1—table 8* and *Figure 5B*). Similar effects were found in terms of numbers of penetrated sperm per ZP (*Figure 5A and B*).

ZPs that had not been exposed to bovine OVGP1 all showed attached sperm of the three species; bovine sperm specific binding most, followed by mouse sperm and then human sperm (*Figure 5A*). However, when ZPs had been exposed to bovine OVGP1, only bovine sperm were found to bind to bovine ZPs (*Figure 5B*).

Since, under in vivo conditions, sperm comes into contact with the female's oviductal tract, and consequently with OVGP1, we investigated the potential effect of this oviductal fluid protein on the behavior of the semen sample when exposed to ZPs. To this end, we conducted the EZPT using ZPs from ovarian bovine oocytes that were either treated with bOVGP1 or left untreated (for 30 min). In this experiment, we used both homologous (bovine) and heterologous (murine) sperm. Additionally, capacitated sperm was either treated or left untreated for 30 min with bovine or murine OVGP1. As shown in *Appendix 1—table 9*, we found that treating semen samples with bovine or murine OVGP1

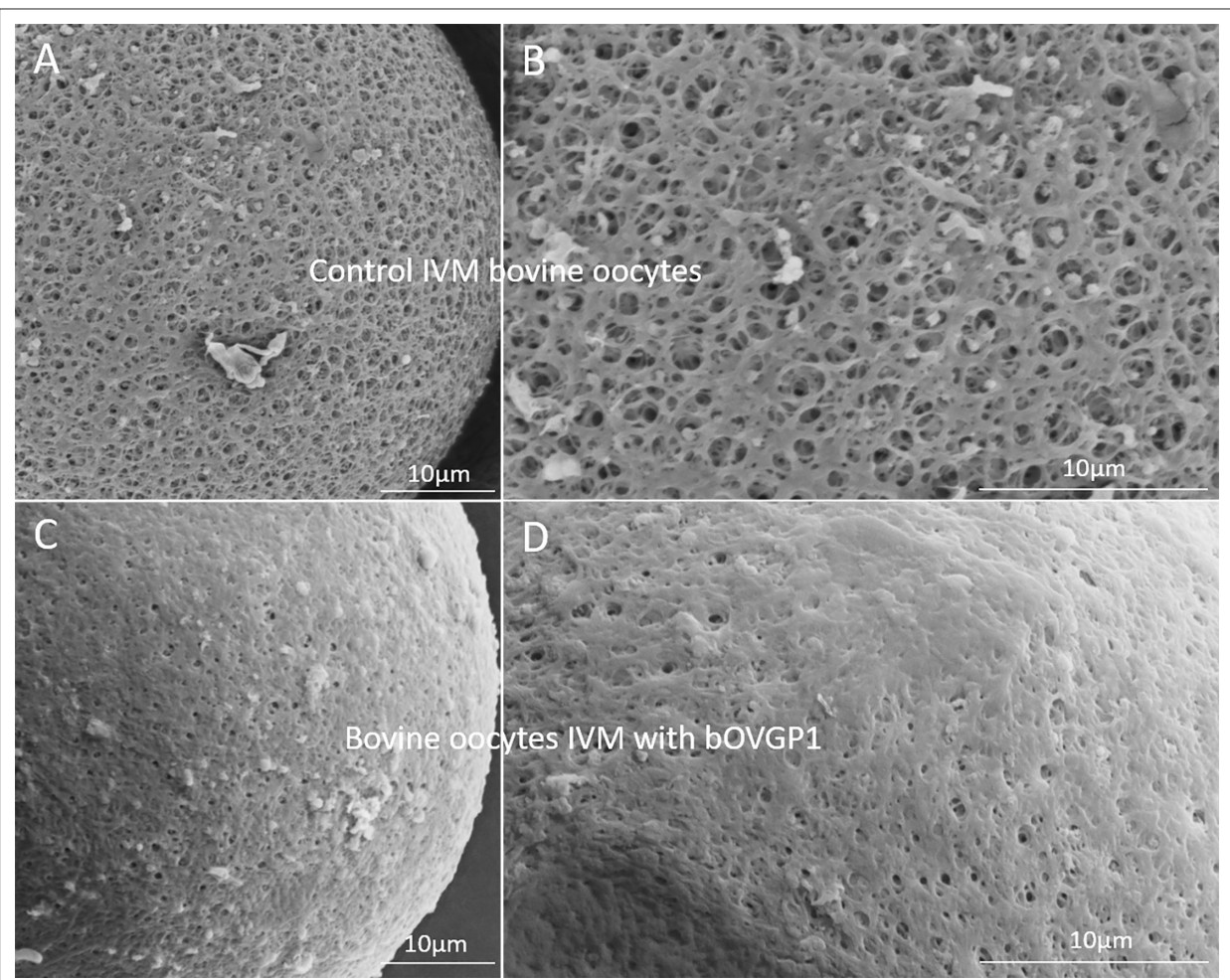

**Figure 6.** Scanning electron micrographs (SEM) of the outer surface of the bovine ZP treated or not with OVGP1. The ZP of an in vitro matured (IVM) bovine oocyte: Magnification x2000 (**A**) and x4000 (**B**); and bovine oocytes IVM in the presence of bovine OVGP1 (bOVGP1): Magnification x2000 (**C**) and x4000 (**D**). High magnification reveals the ultrastructural characteristics of the ZP's pores on the IVM oocytes without bOVGP1 (**B**) and with bOVGP1 (**D**). Scale bars = 10 μm.

The online version of this article includes the following figure supplement(s) for figure 6:

**Figure supplement 1.** Scanning electron micrographs (SEM) of the outer surface of the bovine ZP.

did not affect their ability to penetrate or bind to ZPs, yielding results similar to those observed in previous experiments (*Figure 5A and B* and *Appendix 1—table 8*).

## Observation by scanning electron microscopy (SEM) of the effect of oviductal fluid and homologous or heterologous OVGP1 on the structure of the bovine zona pellucida

After in vitro maturation (IVM), oocytes displayed a porous ZP structure characterized by fine-meshed reticular pores; the pores were circular or elliptical, arbitrarily distributed, and accompanied by deep pore holes (*Figure 6A and B*). Upon exposure to bovine OVGP1 for 30 min, bOVGP1 deposition on the outer ZP surface was observed (*Figure 6C and D*). Pores were larger and more numerous for IVM oocytes that had not been in contact with bOVGP1. A similar effect was observed with bOF (*Figure 6—figure supplement 1A and B*), and with human or murine OVGP1 (*Figure 6—figure supplement 1C*). In this SEM study, we were thus able to visualize the morphological changes occurring on the outer ZP surface of IVM bovine oocytes following their exposure to OVGP1.

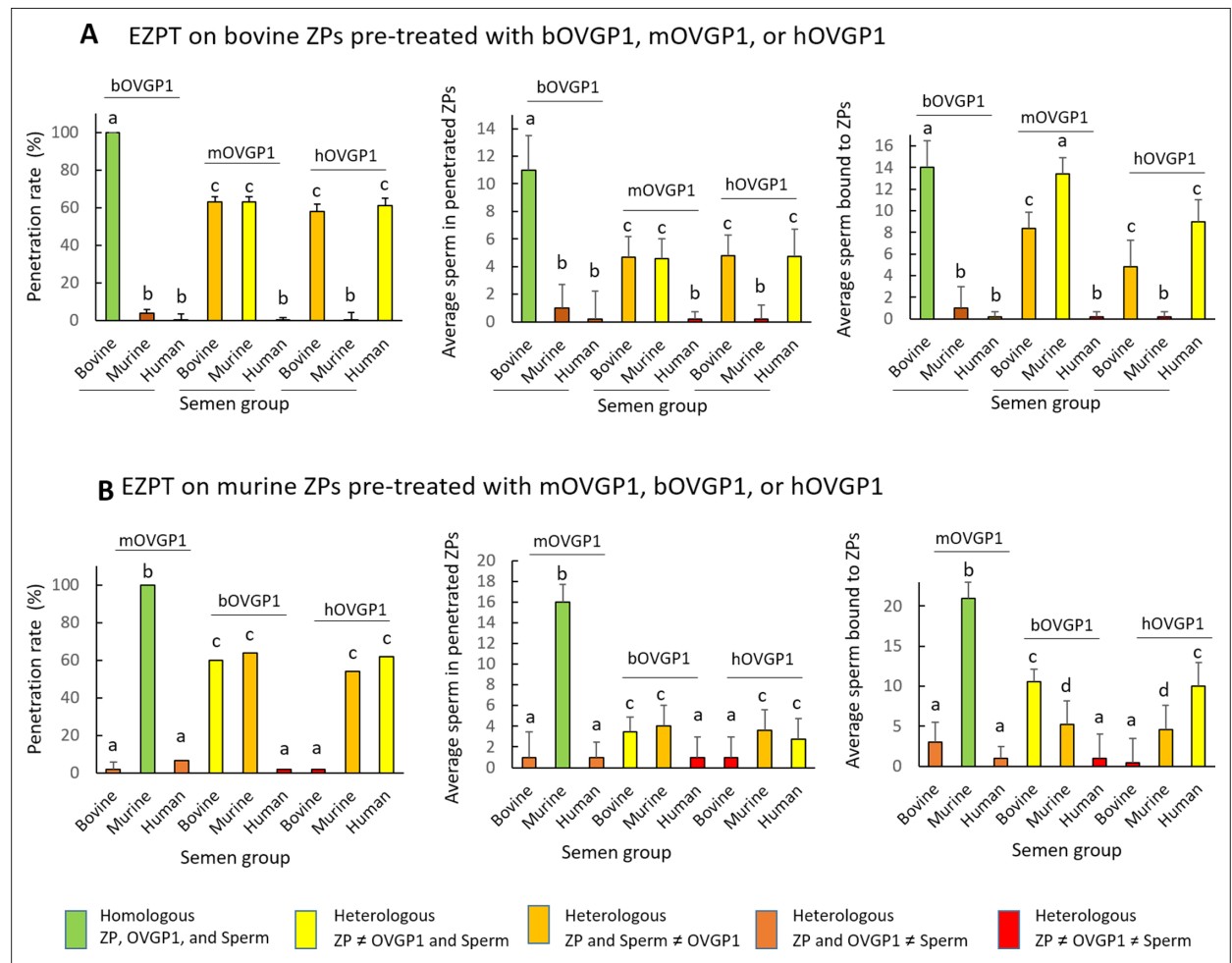

**Figure 7.** Species-specific OVGP1 confers complete species-specificity to the zona pellucida. Experiment using the EZPT to analyze several combinations whereby ZPs from bovine and murine ovarian IVM oocytes were co-incubated with bovine, murine, or human oviductin, and exposed to sperm of all three species. The variables ZP penetration rate, average number of sperm penetrating the ZPs, and average number of sperm bound to ZPs were examined using bovine ZPs (**A**) or murine ZPs (**B**) and combinations of one of the three oviductins with sperm of one of the three species. Only by combining ZP, OVGP1 and sperm of the same species, was species-specific fertilization possible. However, when the same species was matched only with the oviductin and sperm, or only with the ZP and sperm, penetration and binding to the ZP of the spermatozoa also occurred, although in much smaller measure. No penetration was observed when only the ZP and OVGP1 species matched. Different letters above error bars (mean ± SD) indicate significant differences (p<0.05) among groups (ANOVA and Tukey's post hoc test). Numbers of ZPs used are indicated in *Appendix 1—table 10*, *Appendix 1—table 11*.

## Only species-specific OVGP1 confers species-specificity to the zona pellucida

We conducted an experiment to determine whether oviductin from a species different from the origin of the ZP could confer specificity. Through the EZPT several combinations were tested whereby ZPs from bovine or murine ovarian oocytes were co-incubated with bovine, murine, or human OVGP1 and then exposed to sperm of the three species. As shown in *Figure 7A* and *Appendix 1—table 10*, for bovine ZPs and bOVGP1, penetration of the empty ZP was only possible in the case of bovine sperm, such that the entry of sperm of the other two species was completely blocked. For the combination bovine ZP/mOVGP1, human sperm were fully blocked, while some penetration (approximately 60%) by bovine sperm (homologous to the ZP) or murine sperm (homologous to OVGP1) was possible. Similar results were observed for bovine ZP/hOVGP1, sperm homologous to the ZP (bovine) and their pretreatment with human OVGP1, giving rise to partial penetration capacity (60%), indicating that OVGP1 non-homologous to the ZP allow the partial passage of sperm of the ZP's species of origin and of the OVGP1-providing species. Similar penetration results were obtained using murine ZPs in combination with the three oviductin and sperm species (*Figure 7B*, *Appendix 1—table 11*), indicating that results were not specific to bovine ZP.

Results corresponding to mean numbers of penetrated sperm (*Figure 7A and B*, **center graphs**) were consistent with these previous findings reflecting a clear reduction when the OVGP1 source was not homologous to that of the ZP, but the sperm used belonged to one of these two species. When there was no species homology among ZP, OVGP1, or sperm, the mean number of sperm recorded was less than one.

When we examined mean numbers of sperm bound to the ZP (*Figure 7A and B*, **right graphs**), results paralleled those of penetration. Hence, when there was homology among the sperm, ZP, and OVGP1, only sperm of the homologous species were able to bind to the ZP; and when there was only homology between sperm and either OVGP1 or ZP, there was a reduction in the number of sperm found attached to the ZP. Interestingly, species homology between OVGP1 and sperm resulted in more bound sperm than when there was only homology between ZP and sperm, indicating the involvement of OVGP1 in sperm-ZP binding. These differences in sperm binding were, nevertheless, not reflected in the penetration rate.

## Impact of neuraminidase on the EZPT in bovine and murine oocytes pre and post OVGP1 treatment

The enzyme neuraminidase (NMase) cleaves sialic acid residues commonly found in glycoproteins and glycolipids, and is thus able to modify the surface properties of cells and consequently affect their interactions with other molecules or cells. While the involvement of sialic acid in sperm–ZP binding (*Velásquez et al., 2007*; *Pang et al., 2011*) is generally accepted, studies reaching this conclusion have not compared the ZPs of oocytes that have been in contact or not with OVGP1. The aim of this experiment was to examine the influence of NMase treatment on the sperm's ability to bind and to penetrate the ZPs of both bovine and murine oocytes before and after exposure to OVGP1 (*Figure 8*, *Appendix 1—table 12*, *Appendix 1—table 13*).

The results indicate that, regardless of whether ZPs had been or not in contact with OVGP1, after NMase treatment, sperm penetration was completely blocked, and the number of sperm adhering to the ZP was reduced by 70%, suggesting that both nonspecific sperm-ZP binding and species-specific binding occurring after contact with OVGP1 are prevented by NMase treatment, thus impeding sperm penetration (*Figure 8*, *Appendix 1—table 12* and *Appendix 1—table 13*). It is important to note that despite completely blocking penetration, the binding of some sperm to the ZP likely indicates their ability to nonspecifically adhere, which could occur, for example, when the acrosome is damaged.

To confirm that the sialic acids on bOVGP1 are responsible for the species-specific binding of sperm, we conducted an experiment in which only OVGP1 was treated with NMase before coming into contact with the bovine ZP. The treated bOVGP1 was then incubated with the ZP for 30 min, and then we performed the homologous (bovine sperm) or heterologous (mouse sperm) EZPT. In this setup, the ZP retained its sialic acid residues, while OVGP1 did not. As shown in *Figure 9* and *Appendix 1—table 14*, the treatment did not reduce the percentage of penetration when homologous sperm was used, but it halved the number of sperm penetrating the ZP (*Figure 9A*). It is possible that penetration still occurs because there is sialic acid on the ZP, which allows non-species-specific

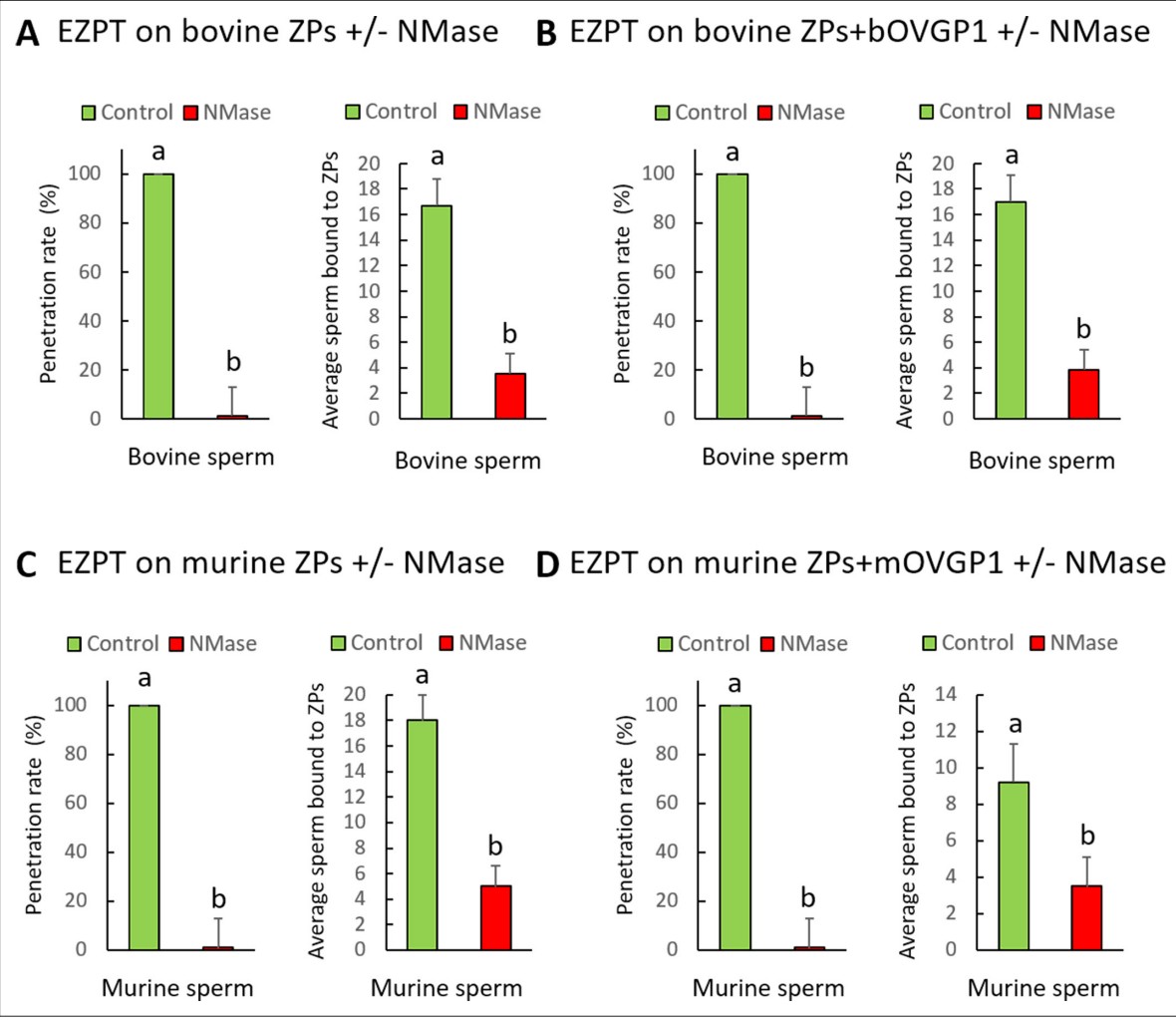

**Figure 8.** Effect of neuraminidase (NMase) treatment of bovine or murine ZPs before and after contact with OVGP1 on sperm penetration. Penetration and sperm binding rates were measured after homologous EZPT with bovine sperm using ZPs not subjected (**A**) or subjected (**B**) to 30 min of incubation with bOVGP1. Penetration and sperm binding rates were measured after homologous EZPT with murine sperm using ZPs not subjected (**C**) or subjected (**D**) to 30 min of incubation with mOVGP1. ZPs of both species were co-incubated with acetate buffer (PH 4.5) at 38 °C for 18 hr in the presence or absence of neuraminidase diluted at 5 UI/mL. Different letters above error bars (mean ± SD) indicate significant differences (p<0.05) among groups (ANOVA and Tukey's post hoc test). Numbers of ZPs used are indicated in **Appendix 1—table 12**, **Appendix 1—table 13**.

sperm binding. However, the number of sperm that penetrate is lower because there is no sialic acid on OVGP1, blocking species-specific binding.

Unlike what happens with bovine sperm, penetration occurred with heterologous mouse sperm in the absence of bOVGP1. However, no penetration was observed in intact bovine ZPs in the presence of non-treated bOVGP1, likely because bOVGP1 and its bovine-specific sialic acids prevent the binding and penetration of mouse sperm. When OVGP1 was treated with NMase, a certain degree of penetration was observed, likely due to the sialic acids present on the ZP, which enable non-species-specific sperm binding (**Figure 9B**). These findings highlight the critical role of sialic acids in sperm binding, with those on the ZP facilitating non-specific binding and those on OVGP1 playing a key role in species-specific binding.

## Discussion

The ZP exhibits different characteristics before and after ovulation and fertilization (**Moros-Nicolás et al., 2021**). The present study provides evidence that both bovine and murine oocytes acquire species-specific fertilization capacity within the oviduct. Oocytes from the ovaries of cows and mice

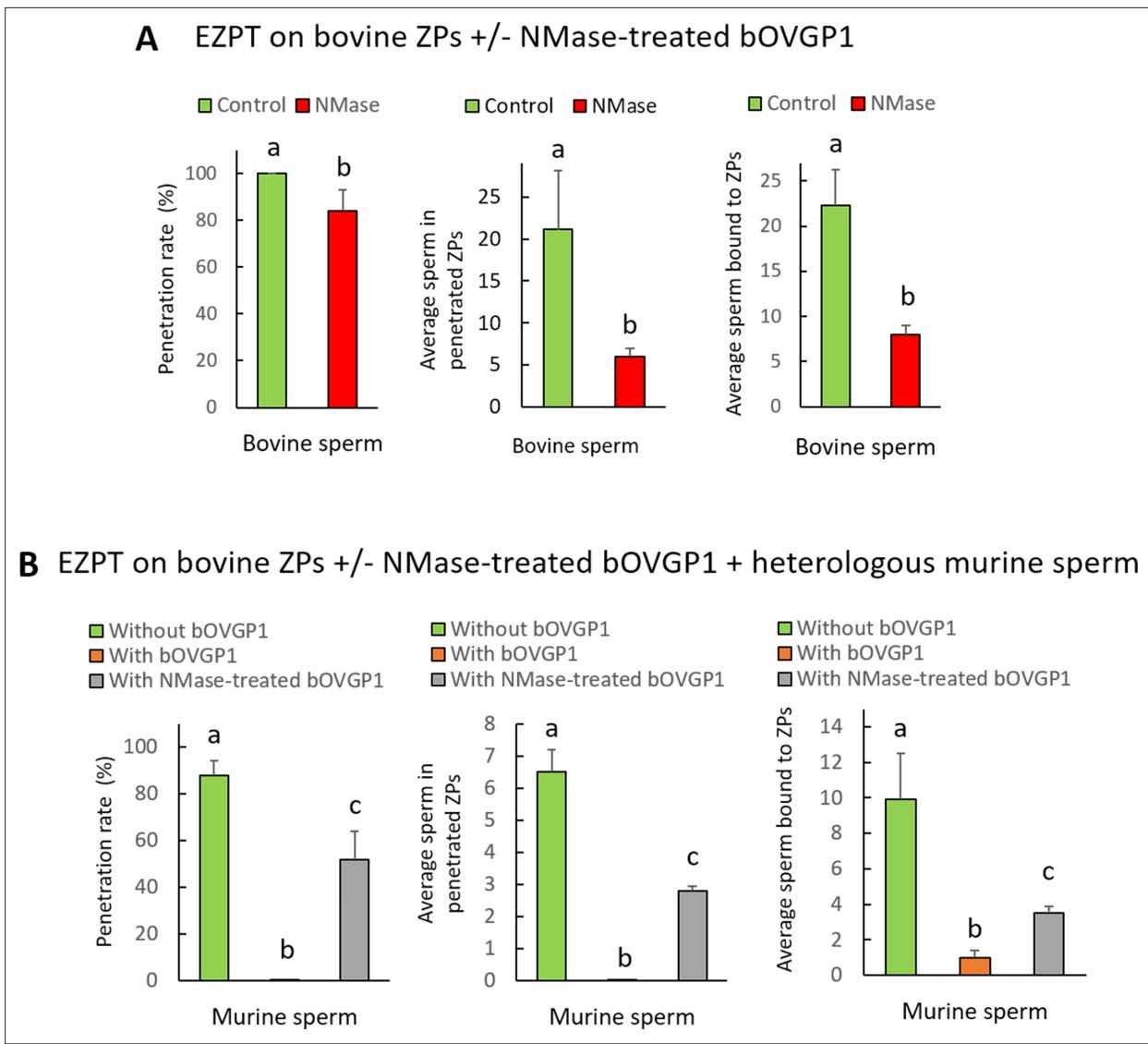

**Figure 9.** Effect of neuraminidase (NMase) treatment of OVGP1 on EZPT using bovine ZPs with homologous or heterologous sperm. (**A**) Penetration rate, average number of sperm within penetrated ZPs, and average of number of sperm bound to ZPs were measured after homologous EZPT with bovine sperm, using ZPs untreated with NMase and bOVGP1 either treated or untreated with NMase. (**B**) Penetration rate, average number of sperm within penetrated ZPs, and average of number of sperm bound to ZPs were measured after homologous (bovine sperm) or heterologous (mouse sperm) EZPT, using ZPs untreated with NMase, either without bOVGP1 or with bOVGP1 treated or untreated with NMase. Numbers of ZPs used are indicated in *Appendix 1—table 14*.

can be fertilized by sperm from distant mammalian species, yet they acquire species-specificity upon contact with oviductal fluid or specific OVGP1 protein (*Figure 10*). Our results unveil two distinct mechanisms for sperm to recognize and penetrate the ZP. One is nonspecific and acts when oocytes come directly from the ovary, while the other is specific and operates when oocytes reach the oviduct and encounter oviductal fluid or its main secreted constituent, OVGP1. In the nonspecific mechanism, carbohydrate residues present in the ZP, particularly those containing sialic acid, are necessary, allowing sperm to penetrate through the ZP's matrix. For the species-restricted mechanism, oligosaccharides chains containing sialic acids from OVGP1 must be recognized by sperm for them to bind and penetrate the ZP. These findings indicate that species-specific OVGP1 determines the species-specificity of the ZP in mammals. Furthermore, our results suggest that heterologous IVF using bovine or murine ovarian oocytes or their ZPs could be an excellent approach for assessing sperm capacitation and fertility in mammalian species where obtaining same-species oocytes is challenging. This

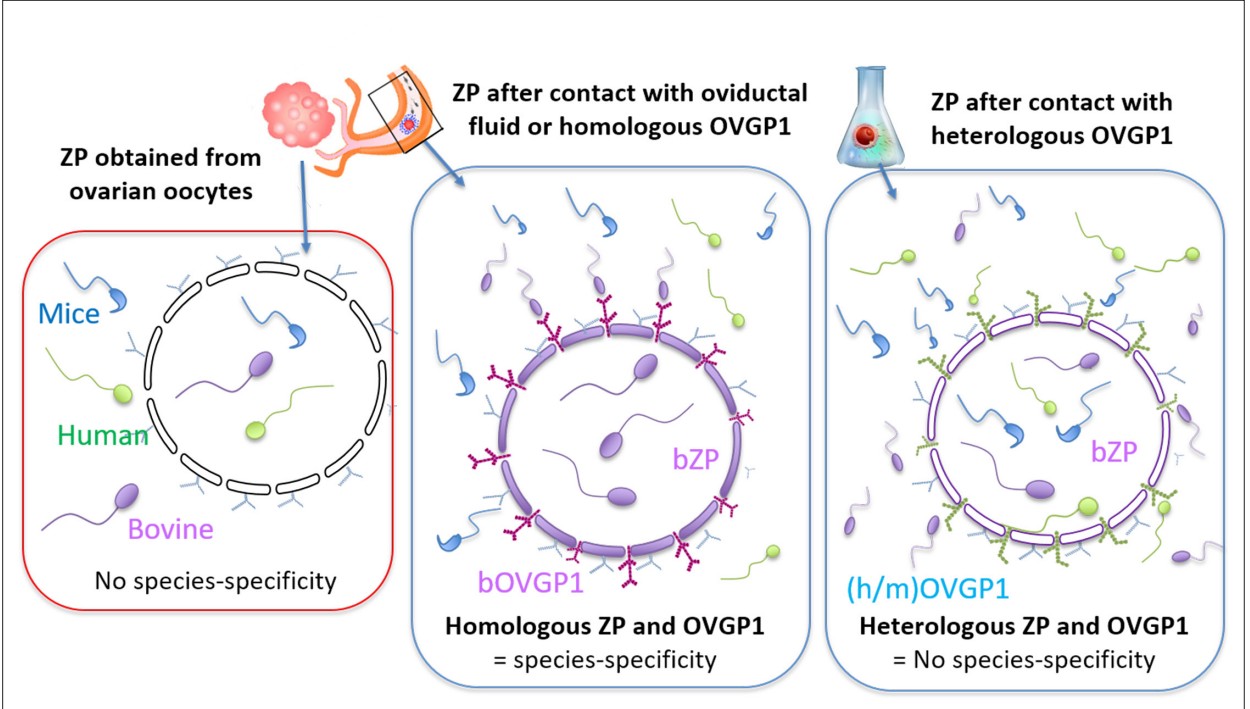

**Figure 10.** Model. Schematic representation of the heterologous fertilization model via EZPT under multiple conditions.The figure illustrates how sperm penetration into the zona pellucida (ZP) occurs non-specifically among various mammalian species when the ZP has not been exposed to homologous oviductal fluid or OVGP1 (left diagram). However, this interaction becomes species-specific when the ZP is incubated with oviductal fluid or OVGP1 from the same species (central diagram) This specificity is lost when OVGP1 is heterologous to the ZP (right diagram).

study contributes to our understanding of reproductive biology and sheds light on the molecular mechanisms driving cross-species fertilization.

Species-specific interactions between gamete membranes in mammals are not always clear-cut. While the exact mechanism of fusion between sperm and oolemma membranes is only partially understood, several molecules such as IZUMO, TMEM95, SPACA6, and SOF1 have been incriminated (*Inoue et al., 2005*; *Lamas-Toranzo et al., 2020*; *Noda et al., 2020*). The EZPT enables heterologous sperm to overcome the oocyte's second barrier, the plasma membrane or oolema. The test known as HEPT, which employs ZP-free oocytes, has shown that the hamster oocyte membrane can fuse with the sperm membrane of many mammalian species but the presence of ZP allows only the penetration of hamstersperm. Our study reveals that bovine and murine ovarian oocytes permit the penetration of both the ZP and oolema of various mammalian orders. This suggests that, prior to reaching the oviduct, a similar mechanism is used across mammalian species for heterologous sperm to recognize and penetrate the ZP and oocyte plasma membrane. However, when the oviduct is reached, some components play a crucial role in inducing species-specific alterations in the ZP, preventing the recognition and/or penetration of the sperm of other species.

During folliculogenesis and transit through the oviduct, the ZP of ovarian oocytes undergoes maturation in preparation for interactions with spermatozoa in the female reproductive tract (*Coy et al., 2008*; *Avilés et al., 2000a*; *Oikawa et al., 1988*). These changes occur when ovulated oocytes are exposed to oviductal fluid, although the exact molecular mechanisms are incompletely understood, it was previously reported changes in carbohydrate composition and incorporation of proteins (*Avilés et al., 2000a*; *Avilés et al., 2000b*). It has been observed that immature oocytes possess a smooth, compact ZP surface with few pores, while IVM oocytes exhibit a net-like porous surface with a rough pattern on the zona matrix (*Báez et al., 2019*; *Familiari et al., 1988*). Here, we observed that IVM ZPs exposed to OVGP1, undergo a complete transformation, oviductin covering most of the ZP and its pores. This suggests that OVGP1 contributes to the maturation of the oviductal ZP by influencing matrix morphology. At the point of fertilization, OVGP1 is found as the main non-serum glycoprotein in oviductal fluid and its relationship with the ZP was noted early on *Araki et al., 1987*. OVGP1 plays a

role in strengthening the ZP, rendering it resistant to proteolytic digestion and sperm penetration (*Coy et al., 2008*), and these processes are regulated by OVGP1's C-terminal region (*Algarra et al., 2016*). The C-terminal region of OVGP1 is the region which mainly differ among species (*Avilés et al., 2010*) and this fact may explain the species-specificity of sperm penetrability. A recently study demonstrates that the OVGP1 gene is essential for normal early embryonic development, as OVGP1-KO hamsters exhibit fertilization failures, abnormal embryos, and low implantation rates. These findings highlight the critical role of the oviductal microenvironment in regulating early embryogenesis; however, they did not analyze the interaction between OVGP1 and ZP (*Yamatoya et al., 2024*).

In mammals, it is accepted that fertilization typically occurs within the ampulla of the oviduct, where sperm bind to the oocyte through interactions with ZP via N- and O-glycoproteins (*Tecle et al., 2019*). Over time, the proposed molecular model explaining sperm-oocyte interactions has evolved. Initially, studies suggested ZP2 and ZP3 acted as primary sperm receptors (*Bleil and Wassarman, 1980*; *Bleil and Wassarman, 1988*; *Bleil and Wassarman, 1990*). Later, it was proposed that sperm-ZP interactions might depend more on the supramolecular structure of the oocyte coat rather than specific ZP subunits (*Rankin et al., 2003*). More recent research points to a nuanced perspective, highlighting a central role for a specific domain of ZP2 in gamete recognition, which physically interacts with complementary molecules on sperm (*Avella et al., 2014*). Despite these advances, the exact roles of ZP and OVGP1 glycosylation and sialic acid residues (Sias) in fertilization are poorly understood. After ovulation, the changes reported in the carbohydrate composition of the ZP (*Moros-Nicolás et al., 2021*; *Velásquez et al., 2007*) are likely induced by the addition of glycoproteins of oviductal origin, as we have seen here with OVGP1. By employing NMase, to remove sialic acid residues from ZP glycoproteins, we here ascertain that OVGP1 plays a pivotal role in the modifications to the carbohydrate composition of the ZP necessary for species-specific sperm fertilization. In conclusion, our results indicate that the oocyte ZP must interact with the contents of the oviductal fluid, specifically with OVGP1 protein, to ensure that only sperm from the same species can fertilize the egg. This determines that the modifications introduced by OVGP1 in the ZP are critical to define the barrier among the different species of mammals. It is noteworthy that this work strengthens the idea that incubating oocytes with oviductal fluid (or OVGP1) can be used to reduce polyspermy in porcine IVF (*Coy et al., 2008*; *Kouba et al., 2000*; *Batista et al., 2016*). Additionally, future studies would be valuable to investigate if ZPs could naturally enhance sperm selection in human ICSI.

## Materials and methods

### Reagents

All media components were purchased from Sigma–Aldrich (St. Louis, MO, USA), except where stated otherwise.

### Sperm collection, cryopreservation, thawing, and capacitation

Human semen samples were obtained from ejaculated sperm samples of three normozoospermic donors with their written informed consent. Approval for the study protocol was obtained from the Hospital Clinico San Carlos Research Ethics Review Committee (Madrid, Spain; Reference number: C.P. IND2022/BIO23646 - C.I. 23/428-E) and the FivCenter fertility clinic (Aravaca, Madrid, Spain) in accordance with the principles of the Declaration of Helsinki. Informed consent for patients was overseen by the Spanish Fertility Society. Donor semen samples were obtained by masturbation after 2–3 days of abstinence and analyzed in the fertility clinic's laboratory. Samples were left to liquefy and then subjected to basic seminogram analysis following WHO guidelines (*Chung et al., 2022*). Sperm concentration and motility assessments were conducted using a Makler counting chamber (BioCare Europe, Rome, Italy).

Semen samples were diluted at a 1:1 ratio (vol:vol) with the freezing medium (TEST Yolk Buffer Fujifilm, Irvine Scientific) and stored for 40 min at 4 °C for equilibration. After this time, the mixture was plunged drop by drop (10 μL) onto a dry ice plate, so that small solid microspheres were formed. These microspheres were then recovered and placed into cryovials for storage in liquid nitrogen at –196 °C. For thawing, the desired pellets were transferred to a 15 mL Falcon tube and kept at 37 °C for 20 min. Next, the samples were washed by diluting 1:1 in human G-IVF PLUS medium (Vitrolife, Igenomix) and centrifuging at 350 × *g* for 10 min. After discarding the supernatant, 200–220 μL of

G-IVF PLUS medium was added to the sediment and the sample stored at 38.5 °C in a 5% $CO_2$ atmosphere for 45–60 min. After this time, 90 µL of the upper culture medium layer was removed using a sterile pipette. This aliquot was ready for use following standard swim-up procedures.

Sperm from B6CBAF1 (C57BL/6xCBA) male mice (aged 8–20 weeks) were used for the heterologous and homologous IVF experiments. The animals were maintained under conditions of 14 hr of light and 10 hr of darkness, with no food or water restrictions. Male mice were individually isolated for a period of 2 weeks before sperm collection. Euthanasia was performed by cervical dislocation. Each cauda epididymis was placed in M2 (M2 medium, Sigma-Aldrich M7167) to wash them out and remove the artery of the vas deferens, followed by washing out with HTF medium (human tubal fluid; 2.04 mM CaCl2, 101.6 mM NaCl, 4.69 mM KCl, 0.37 mM KH2PO4, 0.2 mM MgSO4, 21.4 mM sodium lactate, 0.33 mM sodium pyruvate, 2.78 mM glucose, 25 mM NaHCO3, 100 U/mL penicillin, 50 µg/mL streptomycin and 0.001% (w/v) phenol red); supplemented with 1% (w/v) BSA. Spermatozoa were collected by gently squeezing the dissected vas deferens and cauda epididymis and were then incubated to swim-out in a 500 µL droplet of HTF medium, which supports capacitation, for 30 min at 37 °C in the 5% $CO_2$ incubator. The concentration of the sperm sample was determined using a Thoma cell counting chamber. Sperm motility quality was assessed following placement of 6 µL of sperm suspension in a Mackler chamber on the stage of a microscope heated to 37 °C (Nikon Eclipse E400). Fertilization was conducted in a final concentration of $2 \times 10^5$ capacitated spermatozoa per mL (*Hourcade et al., 2010*). All studies involving mice were approved by the Ethics Committee on Animal Experimentation of the INIA (Madrid, Spain) and were registered with the Dirección General de Agricultura y Ganadería of the Comunidad de Madrid (Spain) under PROEX 137.2/21.

Cat spermatozoa were collected from the cauda epididymis by modified retrograde flushing following vasectomy (*Monaco et al., 2024*). Animal handling complied with Spanish Animal Protection Regulation RD53/2013, which conforms to European Union Regulation 2010/63. Collected spermatozoa were incubated in 200 µL of medium, and their concentration determined using a Neubauer cell counting chamber. For frozen sperm, 50 µL of spermatozoa suspension ($4 \times 10^6$ spermatozoa/mL) were stored in liquid nitrogen vapor for 2 hr to reach a temperature of 4 °C, and then the samples were frozen and stored in liquid nitrogen at –196 °C or lower. The frozen semen straws were thawed at 37 °C in a water bath for 1 min and then centrifuged for 8 min at $300 \times g$ using a gradient of 1 mL of 40% and 1 mL of 80% BoviPure (Nidacon Laboratories AB, Göthenborg, Sweden) according to the manufacturer's instructions. The sperm pellet was isolated and washed in 3 mL of BoviWash (Nidacon Laboratories AB, Göthenborg, Sweden) by centrifugation at $300 \times g$ for 5 min. The pellet was re-suspended in the remaining 80 µL of BoviWash. Fertilization was carried out in a final concentration of $2 \times 10^6$ capacitated spermatozoa/mL.

## FITC-PNA staining and visualization

The spermatozoa were diluted in PBS to a concentration of $2 \times 10^6$ spermatozoa/ml. Twenty µL of the mixture were placed on a clean glass slide and gently pressed between tissue paper sheets to remove excess liquid. The glass slides were heated to 37 °C until the droplets of the sample dilution in PBS were dry. Once dry, the slides were immersed in cold methanol for 30 s and then left to dry for 15 min at RT. Next, two washes were performed in PBS for 5 min. To carry out the staining process, a standard dilution of 1:10 FITC-PNA (150 µg/mL; L7381, Sigma-Aldrich, Madrid) in Hoechst (0.065 µg/mL; B2883, Sigma-Aldrich) was prepared. Twenty µL of this solution were added to the spermatozoa sample on the glass slide. The stained samples were stored in a dark, humid box for a minimum of 30 min. After this time, the slides were washed in distilled water for 10 min, removing the excess and letting them dry on a heated plate at 37 °C. Once dry, and keeping the slides in darkness, 2 µL of the commercial Fluoromount (F4680, Sigma-Aldrich) mixture were placed on the slide, covered with a coverslip (22×22 mm), and gently pressed between tissue paper sheets to remove excess liquid. After 30 min, the samples were ready for visualization. Spermatozoa were observed under a phase contrast microscope at ×1000 magnification with fluorescence illumination.

## Homologous and heterologous IVF of bovine oocytes

Ovaries from mature heifers were collected at a local slaughterhouse, and immature cumulus–oocyte complexes (COCs) obtained by aspirating follicles of diameter 2–8 mm (*Rizos et al., 2010*). Groups of 50 COCs were cultured in four-well dishes (Nunc) for 24 hr at 38.5 °C under 5% $CO_2$ in air, using

500 µL of maturation medium. This medium was composed of TCM 199 supplemented with 10% fetal calf serum (FCS) and 10 ng/mL of epidermal growth factor (EGF).

For homologous fertilization, frozen/thawed semen (0.25 mL) from a single Asturian Valley bull was treated by density gradient centrifugation (BoviPure, Nidacon International, Sweden), and matured COCs where then co-incubated with selected sperm at a final concentration of $1 \times 10^6$ spermatozoa per mL for 18–22 hr in 500 µL of fertilization medium (*Ruiz-Díaz et al., 2023*). This medium consisted of Tyrode's medium containing 22 mM sodium lactate, 25 mM bicarbonate, 1 mM sodium pyruvate, and 6 mg/mL fatty acid—free BSA supplemented with 10 µg/mL heparin sodium salt (Calbiochem). Incubations took place in four-well dishes as groups of 50 COCs per well at 38.5 °C under an atmosphere of 5% $CO_2$ in air and maximum humidity.

For heterologous IVF, the bovine mature COCs were co-incubated with spermatozoa from other species (cat, human, and mouse). Sperm-egg interactions were assessed through a sperm-ZP binding assay at 2.5 hours post-incubation (hpi). For this purpose, COCs were vortexed for 3 min, fixed in 4% paraformaldehyde for 30 min, washed with cold PBS, and stained with Hoechst 33342 to count the number of sperm that remained bound to the ZP. This was done using a widefield fluorescence microscope with structured illumination (ApoTome, Zeiss) and UV-2E/C excitation: 340–380 nm and emission: 435–485 nm. In addition, pronuclear formation was assessed at 6, 12, 18, and 22 hpi. For this, presumptive zygotes were treated and examined following the procedures explained above for sperm-ZP binding. The times used in this study for hybrid embryo analysis were similar to those used in other incubation studies performed in other species (*Sánchez-Calabuig et al., 2015*; *Cañón-Beltrán et al., 2023*). For each IVF, three replicates were conducted for every experimental group.

## In vitro culture of presumptive zygotes to first cleavage embryos on Day 2

After 22 hr incubation of bovine oocytes with sperm (bull, human, mice, or cat), presumptive zygotes or COCs were denuded of cumulus cells by vortexing for 3 min and then cultured in groups of approximately 20 in 25 µL droplets of synthetic oviductal fluid (SOF; *Lopera-Vasquez et al., 2017*). The SOF contained 4.2 mM sodium lactate, 0.73 mM sodium pyruvate, 30 mL/mL BME amino acids, 10 mL/mL minimum essential medium (MEM) amino acids, and 1 mg/mL phenol red. These cultures were supplemented with 5% FCS and placed under mineral oil in an atmosphere of 5% $CO_2$, 5% $O_2$, and 90% $N_2$, and maximum humidity at 38.5 °C (cow) or 37 °C (human, mouse, and cat). Forty eight hours after the start of incubation, cleavage was assessed under a stereomicroscope by checking for the presence of two cells per embryo.

## Collection of oviductal fluid

Five stage I (Days 1–4) oviducts and another five stage II (Days 5–9) were selected from slaughtered heifers. The stage of the estrous cycle was estimated based on the morphology of ovarian structures, especially the corpus luteum (CL). Briefly, in stage I, the CL is small and red, and the epithelium has not yet covered the point of follicular rupture. In stage II, the CL is somewhat bigger, vascularization is observed around it, the apex is reddish in color, and the epithelium has already covered the rupture point. Each tract was independently deposited in a plastic bag and transported to the laboratory on ice in a polystyrene box within 2 hr of collection. The oviducts were washed with cold water to remove blood traces and then in cold $Ca^{2+}$ and $Mg^{2+}$-free phosphate-buffered saline (PBS-). Tissue surrounding the oviduct was carefully isolated and the oviducts were flushed with 1 mL of cold PBS- from the ampulla to the isthmus. All manipulations were performed at 4 °C. The fluid from the five oviducts was pooled and then centrifuged once at 300× $g$ for 7 min. Subsequently, the supernatant was centrifuged again at 10,000×$g$ for 30 min at 4 °C to remove debris (*Hamdi et al., 2018*). The resulting supernatant was aliquoted and stored at –80 °C.

## Collection of oviductal fluid from mice

Estrous cycle stage was determined in mice by visual examination of the vaginal opening and cytological examination of the vagina smear. Oviducts from four estrous female mice were collected and washed by flushing from the ampulla to the isthmus with 0.2 mL of $Ca^{2+}$ and $Mg^{2+}$-free PBS. Female mice were sacrificed sequentially and oviductal fluid was washed out with the same PBS- and oviducts kept on ice after each flushing. This was followed by storage at –80 °C.

## Preparation of empty zona pellucidae from bovine ovarian oocytes

COCs obtained from ovaries from slaughtered heifers and matured in vitro underwent a thorough washing process before their immersion in 30 mL of Tyrode's medium. The oocytes were then mechanically denuded by vortexing and gentle pipetting. The denuded oocytes were subjected to three washing cycles using either G-IVF PLUS (Vitrolife, Igenomix) or FERT medium, depending on the experimental requirements. Batches of approximately 30 oocytes were then carefully placed in 400 µL droplets of the selected medium and covered with mineral oil. For the removal of cytoplasmic contents, a micromanipulator was used following the procedure outlined in *Figure 2—figure supplement 4*. The oocyte was secured using a glass capillary holder, while an ICSI needle, attached to a piezoelectric manipulator, was used to create a small opening in the ZP. This opening facilitated the penetration of the needle, allowing for the complete aspiration of the oocyte's cytoplasm. The aspiration needle was then retrieved from the ZP to expel the extracted cytoplasm. Following this, the empty ZP underwent two washing cycles using G-IVF PLUS or FERT medium. These prepared zona pellucidae were either frozen at –20 °C for a duration of several months (3–9 months) or directly utilized in the sperm penetration experiments.

## Collection and in vitro maturation of ovarian cumulus-oocyte complexes in mice

Wild-Type female mice B6CBAF1 (C57BL/6xCBA) aged 8–10 weeks were superovulated by intraperitoneal injections of 0.1 mL of PMSG (pregnant mare serum gonadotropin, Folligon, 5 IU, INIA, Madrid, Spain). Fifty hours after PMSG supplementation, the females were euthanized by cervical dislocation, and their ovaries removed. Collected ovaries were cleaned of any connective tissue and placed in M2 handling medium supplemented with 4 mg/mL of bovine serum albumin fraction V. Antral follicles were gently punctured with 30-gauge needles, and COCs with at least three compact cumulus cells layers were collected in handling medium and matured for 16–17 hr in TCM199 supplemented with 10% FCS and 10 ng/mL EGF (epidermal growth factor) (*López-Cardona et al., 2017*) and kept at 37 °C under an atmosphere of 5% CO2 in air with maximum humidity. Matured oocytes were isolated and placed in an appropriate medium; HTF for IVF and M2 for ZP isolation and later protein incubation.

## Isolation of superovulated mouse oocytes

Metaphase II oocytes were collected from the oviducts of 6- to 8-week-old female mice superovulated with 7.5 IU of PMSG, and an equivalent dose of hCG (human chorionic gonadotropin) 48 hr later. Briefly, 14 hr post-chorionic gonadotropin administration, oviducts were removed from the superovulated female mice and placed in a petri dish containing M2 at RT. After washing, collected oviducts were placed in fresh M2 medium, and COCs previously in contact with oviductal fluid were released from the ampulla with the aid of Dumont #55 forceps and washed in fresh M2 medium until cytoplasm removal.

## Preparation of zona pellucidae from mouse oocytes

After obtaining the COCs using the two methods described, cumulus cells were dispersed by 3–5 min of incubation in M2 medium containing 350 IU/mL of hyaluronidase (Sigma-Aldrich H4272). After washing, oocytes were kept in KSOM medium at 37 °C in an atmosphere of 5% $CO_2$ until use.

To obtain empty ZP, everything except ZP (nucleus, organelles, membranes, and cytoplasmic contents of the oocytes) was removed using a micromanipulator, following the procedure outlined in *Figure 2—figure supplement 4*. The protocol used was similar to that employed for the bovine oocytes with the help of a holding pipette and a blunt microinjection pipette. The decumulated oocytes were placed in a drop of M2 medium, and with the assistance of a piezoelectric manipulator, each ZP was penetrated, followed by the removal of the cytoplasm using the blunt microinjection pipette.

## Empty zona penetration test (EZPT) of murine and bovine zona pellucidas

For EZPTs, murine and bovine ZPs were firstly incubated with 30 ng/µL of recombinants OVGP1 (bovine, murine and human) for 30 min in 20 µL drops. Then, ZP were washed out in media to remove the excess of recombinant OVGP1.

In some experiments, the ZPs were exposed to oviductal fluid before the incubation with sperm from different species to analyze sperm penetration and sperm binding to the ZPs. In other experiments, the ZPs were either exposed or not exposed to one of the three recombinants OVGP1 (bovine, murine, and human), and each group was incubated with sperm of one of the three species using a final concentration of $1x10^6$ sperm/mL. Male gametes were then incubated with ZPs for 4 hr in 50 µL of HTF medium for murine ZP or 50 µL of FERT medium for bovine ZPs, both covered with mineral oil in a four-well plate at 38.5 °C in a 5% $CO_2$ maximum humidity atmosphere. EZPT was carried out under the same culture conditions for all groups. A positive control was set up through homologous fertilization with bovine or murine semen for each group.

Also, in some experiments, bovine and murine capacitated sperm samples were incubated with OVGP1 under similar conditions to those used for ZPs (incubated with 30 ng/µL of recombinant OVGP1 (bovine and murine) for 30 min in 50 µL drops). To avoid contamination by OVGP1 present in the medium of the sperm samples, these were diluted in the corresponding medium (FERT medium for bovine, HTF for mice) and centrifuged at 1800 x $g$ for 10 min. The supernatant was discarded, and the samples were resuspended in the same media. This procedure was performed twice for each sample. After that, the sperm concentration was adjusted and used as previously described in the protocol for EZPT. Four hours after incubation, the ZPs were transferred from the 50 µL drop of fertilization medium to a new drop to correctly assess both sperm binding and penetration to the ZP. Once the zonas were in the new drop, the spermatozoa binding or inside every ZP were quantified. In each incubation, at least 30 ZPs were used per group and three repetitions were performed for each experimental group. The exact number of ZPs used per group can be seen in *Appendix 1—table 6*, *Appendix 1—table 8*, *Appendix 1—table 9*, *Appendix 1—table 10*, *Appendix 1—table 11*, *Appendix 1—table 12* and *Appendix 1—table 13*.

## Origins of bovine, murine, and human OVGP1 recombinants

The coding sequences of bovine OVGP1 (NM_001080216.1), an additional leader peptide sequence for mammalian cell expression (MGWSCIILFLVATATGVHS) and the sequence to express six-His residues were cloned into a pcDNA3.1(+) expression vector from GenScript Giotech (Rijswijk, NL) containing a promoter for bacteriophage T7 RNA polymerase. Tagged OVGP1 protein was produced by transient expression using a Vaccinia/T7 system (*Elroy-Stein and Moss, 2001*) in BHK-21 cells (ATCC CCL-10) grown in G-MEM BHK-21 medium (Gibco) supplemented with 5% FBS (Gibco), 10% tryptose phosphate medium (Gibco), 20 mM Hepes (Thermo Fisher Scientific), and supplemented with 2 mM L-glutamine (Lonza), 0.1 µg/mL penicillin, and 0.1 µg/mL streptomycin (Lonza). BHK-21 cells grown in six-well plates were infected with a crude stock of vaccinia virus expressing T7 polymerase (vTF7-3; *Fuerst et al., 1987*) at a multiplicity of infection (m.o.i.) of 5 pfu (plaque—forming unit)/cell in 2% FBS EMEM (Eagle's minimum essential medium) medium. After a 90-min adsorption period, the virus inoculum was removed, cell monolayers were washed with EMEM medium and subsequently transfected with pH224 plasmid. Transfection for each well was performed with a mixture of 2 µg of DNA and 6 µL of Fugene HD reagent (Promega). The DNA/fugene mixture was incubated for 15 min at RT prior to its drop-wise addition to the infected culture. At 3 days post-infection, the cell culture was harvested and cells were sedimented by low speed centrifugation.

Bovine OVGP1 (bOVGP1) recombinant protein was purified using the Amicon Pro Purification System (Merck Millipore Ltd) by combining metal chelation chromatography with an Amicon concentrator membrane. Mammalian cell supernatants were incubated under constant agitation for 2 hr with the Ni-NTA His-Bind resin, previously equilibrated with binding buffer. Then, following the manufacturer's protocol specifications and after washing and elution centrifugation steps, His—Tagged proteins were concentrated in an Amicon Ultra-0.5 device, and eluted in 25 µL of the desired buffer (50 mM Tris-HCl, 500 mM NaCl, 5% glycerol, pH = 8) and stored at –80 °C.

Human (NM_002557.4) and mouse (NM_007696.2) recombinant oviductin proteins were obtained from Origene Technologies Inc (human Cat. No. TP321684, murine Cat. No. Tp521693; Rockville, US).

A sequence expressing FLAG—Tagged epitope proteins (DYKDDDDK) was cloned into an expression vector. The recombinant proteins were produced by transient transfection in HEK293T, obtained from the cell culture supernatant, and captured on an affinity column followed by conventional chromatography steps.

## Western blotting of OVGP1

Proteins were run on an SDS-PAGE gel of 10% acrylamide, loading per well 300–500 ng of total purified commercial protein from Origene, and an equivalent amount of the bovine in-house produced protein (from oviductal fluid were used 10 µL of bovine and 50 µL of murine proteins). Protein ladders used were the Broad Multi Color Pre-Stained Protein Standard (M00624S, GenScript) and All Blue prestained Precision Plus Protein Standard (Bio-Rad). The resultant gel was transferred to a nitrocellulose membrane for immunoblotting following standard procedures. Protein loading was assessed after blotting by Ponceau S Solution (Sigma-Aldrich,P7170). Membranes were blocked with 3% BSA in PBS-T (0.05% Tween in 1 X PBS) and incubated with shaking overnight at 4 °C with anti-OVGP1 rabbit polyclonal antibody (NBP1-76939; 1:750) for OVGP1 (able to bind to recombinant human, murine, and bovine OVGP1 and OVGP1 from oviductal fluids), with anti-Histidine Tag antibody rabbit monoclonal clone RM146 (SAB5600227, Sigma-Aldrich; 1:1000) for recombinant bOVGP1, and with Anti- Flag M2 antibody mouse monoclonal (F1804, Sigma-Aldrich; St. Louis, MO, USA; 1:1000) for recombinant hOVGP1 and mOVGP1. Incubation was continued for 2 hr at RT using as secondary antibodies goat anti-rabbit IgG-HRP (Cat. No. GTX213110-01, GeneTex), or goat anti-mouse mIgGk-HRP (sc.516102, Santa Cruz), both diluted 1:5000. As loading control, the primary antibody monoclonal anti-vinculin rabbit antibody (926–42215, LICORbio) was used. Membranes were developed with Immobilon Forte Western HRP Substrate (Cat. No. WBLUF0100, Millipore). The chemiluminescence signal was digitalized using an ImageQuant LAS 500 chemiluminescence CCD camera (GE Healthcare Life Sciences, USA, 29005063).

The total protein amount from oviductal fluids was determined using the PierceTM Protein Assay Kit. The concentration of OVGP1 in the oviductal fluid was quantified using ImageJ software by comparing the mean gray value of the band in the oviductal fluid to the band in the recombinant protein lane. By establishing this relationship, along with the known concentration of protein amount in the recombinant one and in the total protein amount of oviductal fluid, the concentration of OVGP1 in the oviductal fluid was determined as the average of three western blots.

## OVGP1 immunofluorescence

Zona pellucidae were incubated with OVGP1 proteins from human, bovine, and murine (at final concentration of 20 ng/µL), or with murine oviductal fluid for 30 min at RT. Then, they were fixed in 4% PFA (paraformaldehyde) in PBS 1X1% BSA for 20 min at RT. The zonas were washed out twice with PBS 1X1% BSA and incubated with blocking buffer (5% FBS in PBS 1 X) for 45 min. After blocking, primary antibody was prepared at 1:50 in 20% blocking buffer in PBS 1X1% BSA followed by incubation in a humid atmosphere at 4 °C overnight. ZPs were washed out twice in PBS 1X1% BSA and incubated for 2 hr at RT with the secondary antibody 1:200 (in 20% blocking buffer in PBS 1X1% BSA). After three wash-outs in PBS 1X1% BSA, the ZPs were embedded in Fluoromount Aqueous Mounting Medium (F4680, Sigma-Aldrich) and observed under structured illumination in a Zeiss Axio Observer microscope equipped with ApoTome.2 software. The primary antibodies used were anti-OVGP1, rabbit polyclonal (NBP1-76939) for endogenous OVGP1 and recombinant bOVGP1 and mOVGP1 and Anti- Flag M2 antibody, mouse monoclonal (F1804, Sigma-Aldrich; St. Louis, MO, USA) for recombinant hOVGP1. The secondary antibodies used were Alexa Fluor 647 goat anti-rabbit (H+L; A21245, Invitrogen) and Alexa Fluor 488 goat anti-mouse IgG (H+L) (A11029, Invitrogen; *Navarrete-López et al., 2023*).

## Proteomics identification of murine OVGP1

Oviductal fluid from five female mice was obtained in PBS as previously described, and 50 µL aliquots loaded on an 8% polyacrylamide gel for SDS-PAGE. Some lanes of the gel were blotted onto a nitrocellulose membrane for immunoblotting against OVGP1 antibody, and other lanes were fixed for 30 min in methanol:acetic acid (50:10) and rinsed with distilled water. Fixed bands were stained with Coomassie Brilliant Blue, and 75 kDa and 50 kDa bands (compared to the lanes immunoblotted

for OVGP1) were cut, destained in acetonitrile:water (ACN:H2O, 1:1), digested with trypsin, and subjected to reverse phase-liquid chromatography RP-LC-MS/MS mass scan analysis (*Santos et al., 2020*). Murine oviductin identification was performed with the PEAKS Studio v11.5 search engine (Bioinformatics Solutions Inc, Waterloo, Ontario, Canada) against the Uniprot *Mus musculus* database using a filter of peptides and proteins with a FDR ≤ 1% (*Xin et al., 2022*).

## Scanning electron microscopy and image processing

Oocytes were denuded as detailed above and prepared for SEM as previously described (*O'Callaghan et al., 2023*) with minor modifications. Once denuded, oocytes were either exposed or not exposed to one of the three recombinants OVGP1. Briefly, oocytes were placed in fixation medium (2.5% glutaraldehyde [v/v] and 0.1 mol/L sodium cacodylate buffer) for 1 hr at 4 °C. They were then washed with 0.1 mol/L sodium cacodylate buffer and kept in the buffer for 1 hr at 4 °C, followed by washing in distilled water for 5 min. Thereafter, the oocytes were dehydrated with increasing concentrations of acetone at RT. After their dehydration, they were $CO_2$-critical point dried in a stainless chamber (Denton Vacuum DCP-1, Denton Vacuum, LLC, Moorestown, NJ, USA), mounted onto aluminum stubs, and coated with platinum sputter-coated with 5.0 nm thin layer (Leica EM ACE 600). Photomicrographs of the ZP of oocytes were taken a FESEM electron microscope. FESEM analyses were performed with a FE-SEM (ApreoS Lovac IML Thermofisher).

## Incubation of ZP, OVGP1, and sperm with neuraminidase (NMase)

Two hundred isolated bovine ZPs and another 200 isolated murine ZPs were treated with *Clostridium perfringens* neuraminidase (type V, Sigma-Aldrich, N2876) in 0.1 M acetate buffer at pH 4.5, as three separate replicates. The enzyme was tested at concentrations of 5 U/mL for an incubation time of 17 hr at 38.5 °C. As control, ZPs were incubated under the same conditions in the absence of NMase. The NMase used shows broad specific activity, cleaving terminal sialic acid residues which are α–2,3- α–2,6- or α–2,8-linked to Gal, GlcNac, or GalNAc oligosaccharides, glycolipids, or glycoproteins (*Bouwstra et al., 1987*).

When bovine and murine sperm were incubated with recombinant OVGP1, the same conditions used for ZP incubation were applied. To prevent OVGP1 from sperm incubation from being included in subsequent IVF procedures, semen samples were washed in Semen Wash Medium (Stroebech Media, Denmark) by discarding the supernatant after 5 min of centrifugation. After washing, the samples were prepared for EZPT with a final concentration of $1 \times 10^6$ sperm/mL.

When bovine OVGP1 protein alone was incubated with NMase, the same conditions used for ZP incubation were applied. OVGP1 was treated with Clostridium perfringens neuraminidase (Type V, Sigma) in 0.1 M acetate buffer at pH 4.5, with three independent replicates for each experiment. The enzyme was tested at a concentration of 5 U/mL with an incubation time of 17 hr at 38.5 °C. As a control, OVGP1 was incubated under the same conditions but without NMase. The NMase used exhibits broad specific activity, cleaving terminal sialic acid residues that are α–2,3-, α–2,6-, or α–2,8-linked to Gal, GlcNAc, or GalNAc oligosaccharides, glycolipids, or glycoproteins (*Bouwstra et al., 1987*). After incubation of OVGP1 with the enzyme, OVGP1 was isolated from the NMase by incubation under constant agitation for 3 hr with the Ni-NTA His-Bind resin, followed by chromatography and concentration through an Amicon membrane, and then treated bOVGP1 was eluted in 25 μL of the desired buffer (50 mM Tris-HCl, 500 mM NaCl, 5% glycerol, pH = 8). After elution, ZPs were normally incubated for 30 min with treated or control bovine OVGP1 to perform EZPT.

## Statistical analysis

Statistical analysis was carried out using the software package SigmaStat (Jandel Scientific, San Rafael, CA). Results are provided as means ± SD. Means were compared and analyzed by repeated measures one-way analysis of variance (ANOVA), followed by Tukey's post hoc test. Significance was set at $p < 0.05$.

## Acknowledgements

This work was supported by grants PID2021-122507OB-I00 awarded to A Gutierrez-Adan and PID2019-111641RB-I00 to D Rizos and PID2021-123091NB-C21 to M Avilés (funded by MCIN/AEI/10.13039/501100011033/and European Union "NextGenerationEU"/PRTR). DF was supported

by a "Doctorados Industriales 2022" fellowship of the Comunidad de Madrid (IND2022/BIO-23646). YCS for a Margarita Salas contract by Universidad Complutense de Madrid (CT18/22), both funded by the European Union – NextGenerationEU program. We greatly appreciate the help with our study provided by the staff of the FivCenter infertility clinic (Aravaca, Madrid, Spain) and the service of microscopy and image analysis of the ACTI of the Universidad de Murcia (Murcia, Spain). Proteomic analysis of the OVGP1 bands was carried out at the CBMSO PROTEIN CHEMISTRY FACILITY of ProteoRed.

## Additional information

### Funding

| Funder | Grant reference number | Author |
|---|---|---|
| Comunidad de Madrid | IND2022/BIO-23646 | Daniel de la Fuente |
| Ministerio de Ciencia, Innovación y Universidades | PID2021-122507OB-I00 | Alfonso Gutierrez-Adan |
| Agencia Estatal de Investigación | PID2021-122507OB-I00 | Alfonso Gutierrez-Adan |
| European Union | PID2021-122507OB-I00 | Alfonso Gutierrez-Adan |
| Ministerio de Ciencia, Innovación y Universidades | PID2019-111641RB-I00 | Dimitrios Rizos |
| Agencia Estatal de Investigación | PID2019-111641RB-I00 | Dimitrios Rizos |
| European Union | PID2019-111641RB-I00 | Dimitrios Rizos |
| Ministerio de Ciencia, Innovación y Universidades | PID2021-123091NB-C21 | Manuel Avilés |
| Agencia Estatal de Investigación | PID2021-123091NB-C21 | Manuel Avilés |
| European Union | PID2021-123091NB-C21 | Manuel Avilés |
| Universidad Complutense de Madrid | CT18/22 | Yulia N Cajas |

The funders had no role in study design, data collection and interpretation, or the decision to submit the work for publication.

### Author contributions

Daniel de la Fuente, Maria Maroto, Data curation, Formal analysis, Investigation, Methodology, Writing – original draft; Yulia N Cajas, Raul Fernandez-Gonzalez, Data curation, Formal analysis, Methodology; Karina Canon-Beltran, Data curation, Formal analysis, Investigation, Methodology; Ana Munoz-Maceda, Data curation, Methodology; Juana M Sanchez-Puig, Rafael Blasco, Paula Cots-Rodríguez, Methodology; Manuel Avilés, Formal analysis, Investigation, Methodology, Writing – original draft; Dimitrios Rizos, Data curation, Formal analysis, Investigation, Visualization, Methodology, Writing – original draft; Alfonso Gutierrez-Adan, Conceptualization, Resources, Formal analysis, Supervision, Funding acquisition, Investigation, Visualization, Writing – original draft, Project administration, Writing - review and editing

### Author ORCIDs

Daniel de la Fuente https://orcid.org/0000-0001-9161-9076
Maria Maroto https://orcid.org/0000-0001-8738-5932
Paula Cots-Rodríguez https://orcid.org/0000-0001-7630-1043
Dimitrios Rizos https://orcid.org/0000-0001-6813-3940
Alfonso Gutierrez-Adan https://orcid.org/0000-0001-9893-9179

## Ethics

Human semen samples were obtained from ejaculated sperm samples of three normozoospermic donors with their written informed consent. Approval for the study protocol was obtained from the Hospital Clinico San Carlos Research Ethics Review Committee (Madrid, Spain) and the FivCenter fertility clinic (Aravaca, Madrid, Spain) in accordance with the principles of the Declaration of Helsinki. Informed consent for patients was overseen by the Spanish Fertility Society.

All studies involving mice were approved by the Ethics Committee on Animal Experimentation of the INIA (Madrid, Spain), and were registered with the Dirección General de Agricultura y Ganadería of the Comunidad de Madrid (Spain) under PROEX 137.2/21.

Reviewer #1 (Public review): https://doi.org/10.7554/eLife.101338.4.sa1
Reviewer #2 (Public review): https://doi.org/10.7554/eLife.101338.4.sa2
Reviewer #3 (Public review): https://doi.org/10.7554/eLife.101338.4.sa3
Author response https://doi.org/10.7554/eLife.101338.4.sa4

## Additional files

### Supplementary files
MDAR checklist

### Data availability
All data generated or analyzed during this study are included in the manuscript and supporting files.

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

# Appendix 1

**Appendix 1—table 1.** Rates of sperm-ZP binding, pronuclear formation, and cleavage after homologous (bovine sperm) and heterologous (human sperm, media A or B) co-incubation with bovine ovarian oocytes recorded at different times post-insemination.

| Semen group (N) | Sperm binding 2.5 hpi, n | Pronuclear formation 6 hpi, n (%) | 12 hpi, n (%) | 18 hpi, n (%) | 22 hpi, n (%) | Cleavage rate 48 hpi, n (%) |
|---|---|---|---|---|---|---|
| Ho (133) | 32 (0.5±0.17) | | | 36 (72.1±7.3)[a] | | 65 (86.0±7.8)[a] |
| HeA (200) | 31 (0.3±0.17) | 32 (34.4±4,7) | 35 (25.5±6.1) | 31 (54.8±7.1)[b] | 30 (56.6±8.1) | 41 (51.2±8.3)[b] |
| HeB (236) | 33 (0.2±0.17) | 36 (30.5±5.4) | 37 (32.1±6.8) | 36 (33.5±7.3)[c] | 38 (34.3±6.6) | 56 (34.4±7.1)[b] |
| Parth (28) | | | | | | 28 (3.3±3.6)[c] |

Ho = homologous IVF with bovine sperm; HeA = heterologous IVF with human sperm using G-IVF PLUS medium; HeB = heterologous IVF with human sperm using Fert medium; Parth = parthenogenic non-fertilized oocytes. Sperm binding was expressed as the average number of spermatozoa that remained bound to the ZP. 'N' refers to the total number of oocytes fertilized per treatment or non-fertilized oocytes (Parth); 'n' refers to the total number of fertilized oocytes/presumptive zygotes or non-fertilized oocytes (Parth) recorded at each time point to determine rates of sperm binding, pronuclear formation, and embryo cleavage. Pronuclear formation was expressed as the percentage of oocytes with two pronuclei at each time point over the number (n) of oocytes examined. Cleavage rate was expressed as the percentage of two-cell embryos over the number (n) of fertilized or non-fertilized (Parth) oocytes. Analysis was conducted by using a one-way ANOVA, followed by Tukey's post hoc test. Significance level was set at $p < 0.05$. Different superscripts (a, b, c) in the same column indicate significant differences ($p < 0.05$; n=3); Values are expressed as the mean ± SD.

**Appendix 1—table 2.** Rates of sperm-ZP binding, pronuclear formation, and cleavage after homologous (bovine sperm) and heterologous (murine sperm) co-incubation with bovine ovarian oocytes at different times post-insemination.

| Semen group (N) | Sperm binding 2.5 hpi, n | Pronuclear formation 18 hpi, n (%) | 20 hpi, n (%) | Cleavage rate 26 hpi, n (%) | 48 hpi, n (%) |
|---|---|---|---|---|---|
| Ho (231) | 46 (0.6±1.0) | 86 (75.8±10.21)[a] | | | 99 (81.8±1.9)[a] |
| He (436) | 47 (0.6±1.0) | 86 (44.3±6.7)[b] | 87 (44.9±8.8) | 85 (44.7±3.6) | 131 (52.6±3.1)[b] |
| Parth (43) | | | | | 43 (4.4±8.7)[c] |

Ho = homologous IVF with bovine sperm; He = heterologous IVF with murine sperm. Parth = parthenogenic non-fertilized oocytes. Sperm binding was expressed as the average number of spermatozoa that remained bound to the ZP. 'N' refers to the total number of oocytes fertilized per treatment or non-fertilized oocytes (Parth); 'n' refers to the total number of fertilized oocytes/presumptive zygotes or non-fertilized oocytes (Parth) recorded at each time point to determine rates of sperm binding, pronuclear formation, and embryo cleavage. Pronuclear formation was expressed as the percentage of oocytes with two pronuclei at each time point over the number (n) of oocytes examined. Cleavage rate was expressed as the percentage of two- cell embryos over the number (n) of fertilized or non-fertilized (Parth) oocytes. Analysis was conducted by using a one-way ANOVA, followed by Tukey's post hoc test. Significance level was set at $p < 0.05$. Different superscripts (a, b, c) in the same column indicate significant differences ($p < 0.05$; n=3); Values are expressed as the mean ± SD.

**Appendix 1—table 3.** Rates of sperm-ZP binding, pronuclear formation, and embryo cleavage after homologous (bovine sperm) and heterologous (cat sperm) co-incubation with bovine ovarian oocytes at different times post-insemination.

| | Sperm binding | Pronuclear formation | | Cleavage |
|---|---|---|---|---|
| Semen group (N) | 2.5 hpi, n | 6 hpi, n (%) | 18 hpi, n (%) | 48 hpi, n (%) |
| Ho (184) | 43 (0.5±0.9) | | 62 (71.0±2.3)[a] | 79 (83.4±1.7)[a] |
| He (259) | 41 (0.2±0.7) | 48 (31.3±17.14) | 58 (34.1±8.8)[b] | 112 (41.3±2.1)[b] |
| Parth (44) | | | | 44 (2.2±6.6)[c] |

Ho = homologous IVF with bovine sperm; He = heterologous IVF with cat sperm using Tyrode's medium; Parth = parthenogenic non-fertilized oocytes. Sperm binding was expressed as the average number of spermatozoa that remained bound to the ZP. 'N' refers to the total number of oocytes fertilized per treatment or non-fertilized oocytes (Parth); 'n' refers to the total number of fertilized oocytes/presumptive zygotes or non-fertilized oocytes (Parth) recorded at each time point to determine rates of sperm-ZP binding, pronuclear formation, and cleavage. Pronuclear formation was expressed as the percentage of oocytes with two pronuclei at each time point over the number (n) of oocytes examined. Cleavage rate was expressed as the percentage of two-cell embryos over the number (n) of fertilized or non-fertilized (Parth) oocytes. Analysis was conducted by using a one-way ANOVA, followed by Tukey's post hoc test. Significance level was set at p<0.05. Different superscripts (a, b, c) in the same column indicate significant differences (p<0.05; n=3); Values are expressed as the mean ± SD.

**Appendix 1—table 4.** Rates of sperm-ZP binding, pronuclear formation, and cleavage after homologous (bovine sperm) and heterologous (human sperm, media A or B) co-incubation with bovine ovarian oocytes (previously incubated with oviductal fluid for 30 min) at different times post-insemination.

| | Sperm binding | Pronuclear formation | | | Cleavage | |
|---|---|---|---|---|---|---|
| Semen group (N) | 2.5 hpi n | 6 hpi n (%) | 12 hpi n (%) | 18 hpi n (%) | 22 hpi n (%) | 48 hpi n (%) |
| Ho (135) | 32 (0.5±0.9)[a] | | | 33 (66.7±3.6)[a] | | 70 (85.7±3.4)[a] |
| HeA (267) | 34 (0.01±0.2)[b] | 38 (0.0±0.0) | 43 (0.0±0.0) | 49 (0.0±0.0)[b] | 44 (0.0±0.0) | 59 (3.3±2.6)[b] |
| HeB (235) | 31 (0.03±0.2)[b] | 36 (0.0±0.0) | 34 (0.0±0.0) | 36 (0.0±0.0)[b] | 36 (0.0±0.0) | 62 (3.1±4.5)[b] |
| Parth (46) | | | | | | 28 (1.8±4.1)[b] |

Ho = homologous IVF with bovine sperm; HeA = heterologous IVF with human sperm in IVF-G medium; HeB = heterologous IVF with human sperm in Fert medium; Parth = parthenogenic non-fertilized oocytes. 'N' refers to the total number of oocytes fertilized per treatment or non-fertilized oocytes (Parth); 'n' refers to the total number of fertilized oocytes/presumptive zygotes or non-fertilized oocytes (Parth) recorded at each time point to determine rates of sperm-ZP binding, pronuclear formation and cleavage. Sperm binding was expressed as the average number of spermatozoa that remained bound to the ZP. Pronuclear formation was expressed as the percentage of oocytes with two pronuclei at each time point over the number (n) of oocytes examined. Cleavage rate was expressed as the percentage of two-cell embryos over the number (n) of fertilized or non fertilized (Parth) oocytes. Analysis was conducted by using a one-way ANOVA, followed by Tukey's post hoc test. Different superscripts (a, b) in the same column indicate significant differences (p<0.05; n=3). Values are expressed as the mean ± SD.

**Appendix 1—table 5.** Rates of sperm penetration and binding to the murine ZP after the homologous (murine sperm) and heterologous (bovine sperm) EZPT using ZPs prepared from oocytes from murine ovaries (HoA, HeA) and from the oviducts of superovulated female mice (HoB, HeB).

| Group N | Sperm | ZPs from murine oocytes | TMC | Penetrated ZPs (%) | Sperm penetration inside ZP (mean) | Max. no. of sperm inside a single ZP | ZP with bound sperm (%) | Sperm bound to ZPs (mean) | Max. no. of sperm bound to a single ZP |
|---|---|---|---|---|---|---|---|---|---|
| HoA 56 | Murine | Ovary | 8.76±1.07 | 56 (100±0)[a] | 848 (15.08±1.19)[a] | 29 ± 5.50[a] | 56 (100±0)[a] | 1042 (19.02±2.21)[a] | 32 ± 2.06[a] |
| HeAb 97 | Bovine | Ovary | 3.67±0.19 | 56 (57.85±5.45)[b] | 167 (2.98±0.02)[b] | 10±1.4[b] | 91 (93.87±2.65)[b] | 303 (3.33±0.52)[b] | 9±1.42[b] |
| HeAh 85 | Human | Ovary | 6.12±0.17 | 60 (70.83±5.89)[b] | 177 (2.95±0,07)[b] | 9±0.71[b] | 79 (92.92±0.59)[b] | 263 (3.34±0.35)[b] | 11±1.41[b] |
| HoB 37 | Murine | Oviduct | 8.76±1.07 | 37 (100±0)[a] | 423 (23.86±0.79)[a] | 18 ± 4.24[a] | 37 (100±0)[a] | 645 (10.02±1.81)[a] | 28 ± 2.83[a] |
| HeBb 131 | Bovine | Oviduct | 3.67±0.19 | 3 (6.08±1.36)[c] | 3 (1±0)[c] | 1 ± 0[c] | 21 (21.46±2.06)[c] | 62 (2.96±0.33)[c] | 4 ± 0.71[c] |
| HeBh 65 | Human | Oviduct | 6.12±0.17 | 3 (4.72±0,39)[c] | 3 (0.05±0.01)[c] | 1 ± 0[c] | 14 (22.50±3.54)[c] | 39 (2.70±0.42)[c] | 5 ± 2.83[c] |

HoA = homologous EZPT with murine sperm (oocytes obtained from ovary). HeAb = heterologous EZPT with bovine sperm (oocytes obtained from ovary). HeAh = heterologous EZPT with human sperm (oocytes obtained from ovary). HoB = homologous EZPT with murine sperm (oocytes obtained from the oviduct of SOV females). HeBb = heterologous EZPT with bovine sperm (oocytes obtained from the oviduct of SOV females). HeBh = heterologous EZPT with human sperm (oocytes obtained from the oviduct of SOV females). TMC = total motile sperm count after capacitation calculated by multiplying the volume by concentration (million sperm/mL) and by the motility (%) after capacitation. Penetrated ZPs refers to the mean number of ZPs penetrated by at least one sperm. ZPs with bound sperm was calculated as the mean number of ZP with, at least, one bound sperm. Sperm penetration was taken as the total number of sperm observed inside the zonas. Bound sperm was expressed as the mean number of sperm that remained bound to the ZP. 'N' refers to the total number of ZPs fertilized per treatment. Analysis was conducted by using a one-way ANOVA, followed by Tukey's post hoc test. Different superscripts (a, b, c) in the same column indicate significant differences (p<0.05; n=3). Values are expressed as the mean ± SD.

**Appendix 1—table 6.** Rates of sperm penetration and binding to the bovine ZP after homologous (bovine sperm) and heterologous (human or murine sperm) EZPT using bovine ZPs co-incubated without bovine oviductal fluid (HoA, HeA) or with bovine oviductal fluid (HoB, HeB).

| Group N | Sperm | Oviductal fluid | TMC | Penetrated ZPs (%) | Sperm penetration inside ZP (mean) | Max. no. of sperm inside a single ZP | ZP with bound sperm (%) | Sperm bound to ZPs (mean) | Max. no. of sperm bound to a single ZP |
|---|---|---|---|---|---|---|---|---|---|
| HoA 90 | Bovine | - | 4.16±0.27 | 90 (100±0)[a] | 853 (9.13±1.23)[a] | 30 ± 1.41[a] | 90 (100±0)[a] | 830 (9.27±0.58)[a] | 23 ± 1.12[a] |
| HeAh 406 | Human | - | 9.14±3.98 | 283 (69.7±12.1)[b] | 1588 (5.61±2.83)[b] | 23±4.2[b] | 96 (23.65±8.59)[b] | 188 (1.49±0.24)[b] | 6±1.27[b] |
| HeAm 83 | Murine | - | 8.76±1.07 | 80 (96.25±5.3)[a] | 300 (3.73±0.31)[C] | 21±2.12[b] | 83 (100±0)[a] | 490 (5.92±0.81)[c] | 20 ± 3.54[a] |
| HoB 56 | Bovine | + | 4.16±0.27 | 56 (100±0)[a] | 585 (10.52±0.56)[a] | 24±2.82[b] | 56 (100±0)[a] | 574 (10.02±1.81)[a] | 23 ± 2.12[a] |
| HeBh 131 | Human | + | 9.14±3.98 | 0 (0±0)[c] | 0 (0±0)[c] | 0 ± 0[c] | 10 (7.42±2.53)[c] | 10 (1±0)[d] | 1 ± 0[c] |
| HeBm 48 | Murine | + | 9.71±0.61 | 4 (0.08±0.005)[c] | 4 (1±0)[c] | 1 ± 0[c] | 20 (41.74±2.46)[b] | 45 (0.94±0.08)[d] | 5±0.71[b] |

HoA = homologous EZPT with bovine sperm. HeAh = heterologous EZPT with human sperm. HeAm = heterologous EZPT with murine sperm. HoB = homologous EZPT with bovine sperm (ZPs were co-incubated with bovine oviductal fluid for 30 min). HeBh = heterologous EZPT with human sperm (ZPs were co-incubated with bovine oviductal fluid for 30 min). HeBm = heterologous EZPT with murine sperm (ZPs were co-incubated with bovine oviductal fluid for 30 min). TMC = total motile sperm count after capacitation calculated by multiplying the volume by concentration (million sperm/mL) and by the motility (%) after capacitation. Penetrated ZPs refers to the mean number of ZPs penetrated by at least one sperm. ZPs with bound sperm was calculated as the mean number of ZPs with, at least, one bound sperm. Sperm penetration was taken as the total number of sperm observed inside the zonas. Bound sperm was expressed as the mean number of sperm that remained bound to the ZP. 'N' refers to the total number of ZPs fertilized per treatment. Analysis was conducted by using a one-way ANOVA, followed by Tukey's post hoc test. Different superscripts (a, b, c, d) in the same column indicate significant differences (p<0.05; n=3). Values are expressed as the mean ± SD.

**Appendix 1—table 7.** O-glycosylation prediction sites in human, mouse and bull OVGP1. Identification and location of the GalNAc—Type O-glycosylation sites with the online server NetOGlyc 4.0 (*Steentoft et al., 2013*).

| HOMO | Start | End | Score | Comment | MUS | Start | End | Score | Comment | BOS | Start | End | Score | Comment |
|---|---|---|---|---|---|---|---|---|---|---|---|---|---|---|
| HOMO | 29 | 29 | 0.148743 | | MUS | 17 | 17 | 0.0664484 | | BOS | 2 | 2 | 0.477639 | |
| HOMO | 34 | 34 | 0.513945 | + | MUS | 20 | 20 | 0.0136154 | + | BOS | 33 | 33 | 0.0141907 | |
| HOMO | 40 | 40 | 0.133798 | | MUS | 29 | 29 | 0.13254 | | BOS | 36 | 36 | 0.0521786 | |
| HOMO | 52 | 52 | 0.0144943 | | MUS | 34 | 34 | 0.5 | + | BOS | 44 | 44 | 0.211234 | |
| HOMO | 60 | 60 | 0.0428766 | | MUS | 40 | 40 | 0.153781 | | BOS | 48 | 48 | 0.201186 | |
| HOMO | 93 | 93 | 0.0158923 | | MUS | 52 | 52 | 0.0118199 | | BOS | 49 | 49 | 0.283351 | |
| HOMO | 96 | 96 | 0.0261905 | | MUS | 60 | 60 | 0.0436814 | | BOS | 52 | 52 | 0.124074 | |
| HOMO | 104 | 104 | 0.216291 | | MUS | 62 | 62 | 0.0894996 | | BOS | 53 | 53 | 0.100269 | |
| HOMO | 105 | 105 | 0.252093 | | MUS | 93 | 93 | 0.0153551 | | BOS | 55 | 55 | 0.207505 | |
| HOMO | 108 | 108 | 0.14834 | | MUS | 96 | 96 | 0.0240849 | | BOS | 62 | 62 | 0.226985 | |
| HOMO | 109 | 109 | 0.315057 | | MUS | 104 | 104 | 0.143094 | | BOS | 63 | 63 | 0.0478178 | |
| HOMO | 112 | 112 | 0.180764 | | MUS | 105 | 105 | 0.199945 | | BOS | 70 | 70 | 0.0242045 | |
| HOMO | 113 | 113 | 0.114855 | | MUS | 108 | 108 | 0.111836 | | BOS | 88 | 88 | 0.081231 | |
| HOMO | 123 | 123 | 0.030659 | | MUS | 112 | 112 | 0.0957522 | | BOS | 95 | 95 | 0.0428404 | |
| HOMO | 126 | 126 | 0.0314989 | | MUS | 113 | 113 | 0.100753 | | BOS | 114 | 114 | 0.078206 | |
| HOMO | 130 | 130 | 0.0220408 | | MUS | 123 | 123 | 0.00798697 | | BOS | 122 | 122 | 0.240522 | |
| HOMO | 148 | 148 | 0.0445812 | | MUS | 126 | 126 | 0.0128272 | | BOS | 126 | 126 | 0.474913 | |
| HOMO | 155 | 155 | 0.0372563 | | MUS | 148 | 148 | 0.0325529 | | BOS | 149 | 149 | 0.0380635 | |
| HOMO | 174 | 174 | 0.0488918 | | MUS | 174 | 174 | 0.0484669 | | BOS | 152 | 152 | 0.00857951 | |
| HOMO | 182 | 182 | 0.160501 | | MUS | 182 | 182 | 0.166503 | | BOS | 158 | 158 | 0.0631895 | |
| HOMO | 186 | 186 | 0.462315 | | MUS | 186 | 186 | 0.388758 | | BOS | 163 | 163 | 0.275655 | |
| HOMO | 194 | 194 | 0.11187 | | MUS | 190 | 190 | 0.0693216 | | BOS | 167 | 167 | 0.487753 | |
| HOMO | 195 | 195 | 0.0571954 | | MUS | 194 | 194 | 0.0973003 | | BOS | 171 | 171 | 0.482996 | |
| HOMO | 212 | 212 | 0.00914751 | | MUS | 195 | 195 | 0.057626 | | BOS | 178 | 178 | 0.0814828 | |

*Appendix 1—table 7 continued on next page*

*Appendix 1—table 7 continued*

| | HOMO Start | End | Score | Comment | MUS Start | End | Score | Comment | BOS Start | End | Score | Comment |
|---|---|---|---|---|---|---|---|---|---|---|---|---|
| HOMO | 218 | 218 | 0.0565867 | | 212 | 212 | 0.00829588 | | 179 | 179 | 0.348292 | |
| HOMO | 223 | 223 | 0.195033 | | 218 | 218 | 0.047986 | | 202 | 202 | 0.139565 | |
| HOMO | 227 | 227 | 0.375705 | | 223 | 223 | 0.18869 | | 206 | 206 | 0.199712 | |
| HOMO | 231 | 231 | 0.310035 | | 227 | 227 | 0.246007 | | 213 | 213 | 0.5 | + |
| HOMO | 238 | 238 | 0.115751 | | 231 | 231 | 0.246397 | | 226 | 226 | 0.696097 | + |
| HOMO | 239 | 239 | 0.335736 | | 236 | 236 | 0.17176 | | 231 | 231 | 0.428867 | |
| HOMO | 253 | 253 | 0.0093353 | | 238 | 238 | 0.102913 | | 276 | 276 | 0.0105995 | |
| HOMO | 262 | 262 | 0.09816 | | 239 | 239 | 0.289418 | | 296 | 296 | 0.0180258 | |
| HOMO | 266 | 266 | 0.142253 | | 251 | 251 | 0.029901 | | 309 | 309 | 0.0717533 | |
| HOMO | 273 | 273 | 0.4854 | | 262 | 262 | 0.133467 | | 317 | 317 | 0.183042 | |
| HOMO | 286 | 286 | 0.829831 | + | 273 | 273 | 0.193716 | | 328 | 328 | 0.222882 | |
| HOMO | 291 | 291 | 0.174597 | | 279 | 279 | 0.662371 | + | 329 | 329 | 0.779465 | + |
| HOMO | 304 | 304 | 0.0267057 | | 281 | 281 | 0.959249 | + | 330 | 330 | 0.813643 | + |
| HOMO | 336 | 336 | 0.0314593 | | 286 | 286 | 0.791757 | + | 332 | 332 | 0.728598 | + |
| HOMO | 338 | 338 | 0.0119623 | | 291 | 291 | 0.32144 | | 338 | 338 | 0.930032 | + |
| HOMO | 356 | 356 | 0.0239896 | | 294 | 294 | 0.286168 | | 339 | 339 | 0.896458 | + |
| HOMO | 365 | 365 | 0.131311 | | 304 | 304 | 0.048395 | | 343 | 343 | 0.821513 | + |
| HOMO | 369 | 369 | 0.120416 | | 334 | 334 | 0.00966718 | | 344 | 344 | 0.892312 | + |
| HOMO | 388 | 388 | 0.491023 | | 336 | 336 | 0.0104758 | | 352 | 352 | 0.862842 | + |
| HOMO | 389 | 389 | 0.824032 | + | 338 | 338 | 0.0196048 | | 354 | 354 | 0.816318 | + |
| HOMO | 390 | 390 | 0.6421 | + | 356 | 356 | 0.0175912 | | 358 | 358 | 0.767588 | + |
| HOMO | 391 | 391 | 0.739353 | + | 365 | 365 | 0.035171 | | 359 | 359 | 0.60832 | |
| HOMO | 398 | 398 | 0.757269 | + | 385 | 385 | 0.265295 | | 373 | 373 | 0.426953 | |
| HOMO | 399 | 399 | 0.872818 | + | 387 | 387 | 0.549473 | + | 375 | 375 | 0.726137 | + |
| HOMO | 403 | 403 | 0.797847 | + | 389 | 389 | 0.61239 | + | 379 | 379 | 0.817664 | + |

*Appendix 1—table 7 continued on next page*

Appendix 1—table 7 continued

| HOMO | Start | End | Score | Comment | MUS | Start | End | Score | Comment | BOS | Start | End | Score | Comment |
|---|---|---|---|---|---|---|---|---|---|---|---|---|---|---|
| HOMO | 404 | 404 | 0.793866 | + | MUS | 390 | 390 | 0.360483 |  | BOS | 381 | 381 | 0.78998 | + |
| HOMO | 405 | 405 | 0.955482 | + | MUS | 398 | 398 | 0.76942 | + | BOS | 383 | 383 | 0.829896 | + |
| HOMO | 406 | 406 | 0.740014 | + | MUS | 399 | 399 | 0.793433 | + | BOS | 385 | 385 | 0.746896 | + |
| HOMO | 414 | 414 | 0.978317 | + | MUS | 400 | 400 | 0.887109 | + | BOS | 392 | 392 | 0.65328 | + |
| HOMO | 415 | 415 | 0.912469 | + | MUS | 404 | 404 | 0.813503 | + | BOS | 394 | 394 | 0.763928 | + |
| HOMO | 418 | 418 | 0.926281 | + | MUS | 408 | 408 | 0.793101 | + | BOS | 396 | 396 | 0.718114 | + |
| HOMO | 419 | 419 | 0.913712 | + | MUS | 411 | 411 | 0.761464 | + | BOS | 399 | 399 | 0.571659 | + |
| HOMO | 421 | 421 | 0.780885 | + | MUS | 414 | 414 | 0.725498 | + | BOS | 404 | 404 | 0.875147 | + |
| HOMO | 433 | 433 | 0.403671 |  | MUS | 418 | 418 | 0.690982 | + | BOS | 411 | 411 | 0.852652 | + |
| HOMO | 443 | 443 | 0.803539 | + | MUS | 419 | 419 | 0.677789 | + | BOS | 413 | 413 | 0.827119 | + |
| HOMO | 445 | 445 | 0.760132 | + | MUS | 421 | 421 | 0.746303 | + | BOS | 420 | 420 | 0.640815 | + |
| HOMO | 449 | 449 | 0.950001 | + | MUS | 433 | 433 | 0.757383 | + | BOS | 421 | 421 | 0.566064 | + |
| HOMO | 450 | 450 | 0.883589 | + | MUS | 434 | 434 | 0.316116 |  | BOS | 428 | 428 | 0.77777 | + |
| HOMO | 452 | 452 | 0.897374 | + | MUS | 444 | 444 | 0.751447 | + | BOS | 441 | 441 | 0.403374 |  |
| HOMO | 454 | 454 | 0.952252 | + | MUS | 445 | 445 | 0.394782 |  | BOS | 446 | 446 | 0.770962 | + |
| HOMO | 457 | 457 | 0.844377 | + | MUS | 448 | 448 | 0.787216 | + | BOS | 459 | 459 | 0.760108 | + |
| HOMO | 459 | 459 | 0.950975 | + | MUS | 451 | 451 | 0.5 | + | BOS | 460 | 460 | 0.831493 | + |
| HOMO | 464 | 464 | 0.852675 | + | MUS | 453 | 453 | 0.541804 | + | BOS | 462 | 462 | 0.757798 | + |
| HOMO | 471 | 471 | 0.696304 | + | MUS | 457 | 457 | 0.835698 | + | BOS | 466 | 466 | 0.653766 | + |
| HOMO | 474 | 474 | 0.846582 | + | MUS | 459 | 459 | 0.749447 | + | BOS | 468 | 468 | 0.823171 | + |
| HOMO | 478 | 478 | 0.776336 | + | MUS | 466 | 466 | 0.72813 | + | BOS | 475 | 475 | 0.116021 |  |
| HOMO | 480 | 480 | 0.745116 | + | MUS | 471 | 471 | 0.756103 | + | | | | | |
| HOMO | 481 | 481 | 0.794363 | + | MUS | 478 | 478 | 0.539962 | + | | | | | |
| HOMO | 486 | 486 | 0.819105 | + | MUS | 480 | 480 | 0.748887 | + | | | | | |
| HOMO | 488 | 488 | 0.884396 | + | MUS | 481 | 481 | 0.616459 | + | | | | | |

Appendix 1—table 7 continued on next page

*Appendix 1—table 7 continued*

| | Start | End | Score | Comment | | Start | End | Score | Comment | | Start | End | Score | Comment |
|---|---|---|---|---|---|---|---|---|---|---|---|---|---|---|
| HOMO | 495 | 495 | 0.723935 | + | MUS | 486 | 486 | 0.774382 | + | | | | | |
| HOMO | 501 | 501 | 0.820028 | + | MUS | 488 | 488 | 0.664138 | + | | | | | |
| HOMO | 503 | 503 | 0.97727 | + | MUS | 489 | 489 | 0.858047 | + | | | | | |
| HOMO | 504 | 504 | 0.886534 | + | MUS | 490 | 490 | 0.843136 | + | | | | | |
| HOMO | 508 | 508 | 0.780642 | + | MUS | 493 | 493 | 0.871285 | + | | | | | |
| HOMO | 510 | 510 | 0.739161 | + | MUS | 495 | 495 | 0.77954 | + | | | | | |
| HOMO | 511 | 511 | 0.895653 | + | MUS | 496 | 496 | 0.965151 | + | | | | | |
| HOMO | 516 | 516 | 0.863073 | + | MUS | 497 | 497 | 0.917272 | + | | | | | |
| HOMO | 518 | 518 | 0.913466 | + | MUS | 500 | 500 | 0.91103 | + | | | | | |
| HOMO | 523 | 523 | 0.767302 | + | MUS | 502 | 502 | 0.736343 | + | | | | | |
| HOMO | 525 | 525 | 0.753387 | + | MUS | 503 | 503 | 0.961134 | + | | | | | |
| HOMO | 531 | 531 | 0.808482 | + | MUS | 504 | 504 | 0.901609 | + | | | | | |
| HOMO | 533 | 533 | 0.898742 | + | MUS | 507 | 507 | 0.921979 | + | | | | | |
| HOMO | 536 | 536 | 0.828278 | + | MUS | 509 | 509 | 0.729352 | + | | | | | |
| HOMO | 539 | 539 | 0.891049 | + | MUS | 510 | 510 | 0.950811 | + | | | | | |
| HOMO | 541 | 541 | 0.95368 | + | MUS | 511 | 511 | 0.896131 | + | | | | | |
| HOMO | 545 | 545 | 0.897366 | + | MUS | 514 | 514 | 0.889054 | + | | | | | |
| HOMO | 546 | 546 | 0.894875 | + | MUS | 516 | 516 | 0.727671 | + | | | | | |
| HOMO | 548 | 548 | 0.726574 | + | MUS | 517 | 517 | 0.975857 | + | | | | | |
| HOMO | 554 | 554 | 0.7768 | + | MUS | 518 | 518 | 0.936021 | + | | | | | |
| HOMO | 556 | 556 | 0.938298 | + | MUS | 521 | 521 | 0.906595 | + | | | | | |
| HOMO | 561 | 561 | 0.917442 | + | MUS | 523 | 523 | 0.623161 | + | | | | | |
| HOMO | 575 | 575 | 0.93519 | + | MUS | 524 | 524 | 0.94394 | + | | | | | |
| HOMO | 578 | 578 | 0.910072 | + | MUS | 525 | 525 | 0.898366 | + | | | | | |
| HOMO | 582 | 582 | 0.93418 | + | MUS | 528 | 528 | 0.923861 | + | | | | | |

*Appendix 1—table 7 continued on next page*

*Appendix 1—table 7 continued*

| | Start | End | Score | Comment |
|---|---|---|---|---|
| HOMO | 584 | 584 | 0.85508 | + |
| HOMO | 589 | 589 | 0.855402 | + |
| HOMO | 598 | 598 | 0.640589 | + |
| HOMO | 599 | 599 | 0.929108 | + |
| HOMO | 603 | 603 | 0.806488 | + |
| HOMO | 623 | 623 | 0.774533 | + |
| HOMO | 624 | 624 | 0.894222 | + |
| HOMO | 625 | 625 | 0.890486 | + |
| HOMO | 634 | 634 | 0.826692 | + |
| HOMO | 650 | 650 | 0.733632 | + |
| HOMO | 651 | 651 | 0.915455 | + |
| HOMO | 654 | 654 | 0.858169 | + |
| HOMO | 656 | 656 | 0.926624 | + |
| HOMO | 659 | 659 | 0.87521 | + |
| HOMO | 660 | 660 | 0.942906 | + |
| HOMO | 663 | 663 | 0.864385 | + |
| HOMO | 672 | 672 | 0.582959 | + |

| | Start | End | Score | Comment |
|---|---|---|---|---|
| MUS | 531 | 531 | 0.871652 | + |
| MUS | 535 | 535 | 0.672261 | + |
| MUS | 537 | 537 | 0.382735 | |
| MUS | 542 | 542 | 0.777675 | + |
| MUS | 545 | 545 | 0.908425 | + |
| MUS | 549 | 549 | 0.882743 | + |
| MUS | 551 | 551 | 0.683389 | + |
| MUS | 553 | 553 | 0.925689 | + |
| MUS | 556 | 556 | 0.931487 | + |
| MUS | 558 | 558 | 0.691547 | + |
| MUS | 559 | 559 | 0.971129 | + |
| MUS | 560 | 560 | 0.938113 | + |
| MUS | 563 | 563 | 0.954235 | + |
| MUS | 566 | 566 | 0.967467 | + |
| MUS | 567 | 567 | 0.932682 | + |
| MUS | 570 | 570 | 0.951799 | + |
| MUS | 572 | 572 | 0.839915 | + |
| MUS | 573 | 573 | 0.963663 | + |
| MUS | 574 | 574 | 0.928422 | + |
| MUS | 577 | 577 | 0.899768 | + |
| MUS | 579 | 579 | 0.767341 | + |
| MUS | 580 | 580 | 0.946279 | + |
| MUS | 581 | 581 | 0.89457 | + |
| MUS | 584 | 584 | 0.886062 | + |
| MUS | 586 | 586 | 0.83614 | + |

*Appendix 1—table 7 continued on next page*

*Appendix 1—table 7 continued*

| Start | End | Score | Comment | | Start | End | Score | Comment | | Start | End | Score | Comment |
|---|---|---|---|---|---|---|---|---|---|---|---|---|---|
| | | | | MUS | 587 | 587 | 0.939191 | + | | | | | |
| | | | | MUS | 588 | 588 | 0.905539 | + | | | | | |
| | | | | MUS | 591 | 591 | 0.880483 | + | | | | | |
| | | | | MUS | 593 | 593 | 0.856288 | + | | | | | |
| | | | | MUS | 594 | 594 | 0.959942 | + | | | | | |
| | | | | MUS | 595 | 595 | 0.898068 | + | | | | | |
| | | | | MUS | 598 | 598 | 0.773555 | + | | | | | |
| | | | | MUS | 600 | 600 | 0.838638 | + | | | | | |
| | | | | MUS | 601 | 601 | 0.945883 | + | | | | | |
| | | | | MUS | 602 | 602 | 0.902728 | + | | | | | |
| | | | | MUS | 605 | 605 | 0.900001 | + | | | | | |
| | | | | MUS | 607 | 607 | 0.877611 | + | | | | | |
| | | | | MUS | 608 | 608 | 0.962159 | + | | | | | |
| | | | | MUS | 612 | 612 | 0.909255 | + | | | | | |
| | | | | MUS | 614 | 614 | 0.699823 | + | | | | | |
| | | | | MUS | 615 | 615 | 0.909116 | + | | | | | |
| | | | | MUS | 619 | 619 | 0.763155 | + | | | | | |
| | | | | MUS | 624 | 624 | 0.836727 | + | | | | | |
| | | | | MUS | 631 | 631 | 0.957254 | + | | | | | |
| | | | | MUS | 632 | 632 | 0.5 | + | | | | | |
| | | | | MUS | 634 | 634 | 0.560477 | + | | | | | |
| | | | | MUS | 639 | 639 | 0.867656 | + | | | | | |
| | | | | MUS | 641 | 641 | 0.831486 | + | | | | | |
| | | | | MUS | 643 | 643 | 0.864738 | + | | | | | |
| | | | | MUS | 647 | 647 | 0.510238 | + | | | | | |

*Appendix 1—table 7 continued on next page*

*Appendix 1—table 7 continued*

| Start | End | Score | Comment | Comment | Start | End | Score | Comment | Start | End | Score | Comment |
|---|---|---|---|---|---|---|---|---|---|---|---|---|
| | | | | MUS | 648 | 648 | 0.532835 | + | | | | |
| | | | | MUS | 656 | 656 | 0.70976 | + | | | | |
| | | | | MUS | 658 | 658 | 0.725052 | + | | | | |
| | | | | MUS | 662 | 662 | 0.732086 | + | | | | |
| | | | | MUS | 663 | 663 | 0.682057 | + | | | | |
| | | | | MUS | 667 | 667 | 0.400653 | | | | | |
| | | | | MUS | 674 | 674 | 0.731579 | + | | | | |
| | | | | MUS | 676 | 676 | 0.683573 | + | | | | |
| | | | | MUS | 681 | 681 | 0.684517 | + | | | | |
| | | | | MUS | 683 | 683 | 0.602425 | + | | | | |
| | | | | MUS | 688 | 688 | 0.565729 | + | | | | |
| | | | | MUS | 690 | 690 | 0.315646 | | | | | |
| | | | | MUS | 695 | 695 | 0.476853 | | | | | |
| | | | | MUS | 702 | 702 | 0.0864334 | | | | | |
| | | | | MUS | 704 | 704 | 0.115475 | | | | | |

**Appendix 1—table 8.** Rates of sperm penetration and binding to bovine ZP after homologous (bovine sperm) or heterologous (human and murine sperm) EZPT using bovine ZPs co-incubated without bovine OVGP1 (HoA, HeAh, HeAm) or with bovine OVGP1 (HoB, HeBh, HeBm).

| Group N | Sperm | Bovine OVGP1 | TMC | Penetrated ZPs (%) | Sperm penetration inside ZP (mean) | Max. no. of sperm inside a single ZP | ZP with bound sperm (%) | Sperm bound to ZPs (mean) | Max. no. of sperm bound to a single ZP |
|---|---|---|---|---|---|---|---|---|---|
| HoA 67 | Bovine | - | 4.15±0.61 | 67 (100±0)[a] | 730 (10.89±0.40)[a] | 33±5.59 | 67 (100±0) | 918 (13.70±0.21)[a] | 25±2.45 |
| HeAh 30 | Human | - | 8,81±0.74 | 21 (70.0±10.0)[b] | 182 (8.66±0.68)[b] | 13±2.08 | 10 (33.33±5.77) | 18 (0.60±0.66)[b] | 3±0.58 |
| HeAm 83 | Murine | - | 8.76±1.07 | 80 (96.25±5.3)[c] | 300 (3.75±0.31)[c] | 11±2.12 | 83 (100±0) | 490 (5.90±0.81)[c] | 20±3.54 |
| HoB 70 | Bovine | + | 4.15±0.61 | 69 (98.57±1.36)[a] | 708 (10.26±1.51)[a] | 29±4.30 | 70 (100±0) | 768 (10.97±1.31)[a] | 22±2.38 |
| HeBh 90 | Human | + | 8,81±0.74 | 0 (0±0)[d] | 0 (0±0)[d] | 0±0 | 12 (13.33±3.33) | 12 (0.13±0.03)[d] | 1±0 |
| HeBm 81 | Murine | + | 8.76±1.07 | 3 (3.79±2.08)[d] | 3 (1±0)[d] | 1±0 | 36 (44.31±3.16) | 94 (1.16±0.28)[d] | 6±0.71 |

HoA = homologous EZPT with bovine sperm. HeAh = heterologous EZPT with human sperm. HeAm = heterologous EZPT with murine sperm. HoB = homologous EZPT with bovine sperm (ZPs were co-incubated with bovine OVGP1 for 30 min). HeBh = heterologous EZPT with human sperm (ZPs were co-incubated with bovine OVGP1 for 30 min). HeBm = heterologous EZPT with murine sperm (ZPs were co incubated with bovine OVGP1 for 30 min). TMC = total motile sperm count after capacitation calculated by multiplying the volume by concentration (million sperm/mL) and by the motility (%) after capacitation. Penetrated ZPs refers to the mean number of ZPs penetrated by at least one sperm. ZPs with bound sperm was calculated as the mean number of ZP with, at least, one bound sperm. Sperm penetration was taken as the total number of sperm observed inside the zonas. Bound sperm was expressed as the mean number of sperm that remained bound to the ZP. 'N' refers to the total number of ZPs fertilized per treatment. Analysis was conducted by using a one-way ANOVA, followed by Tukey´s post hoc test. Different superscripts (a, b, c, d) in the same column indicate significant differences ($p < 0.05$; n=3). Values are expressed as the mean ± SD.

**Appendix 1—table 9.** Rates of sperm penetration and binding to bovine ZP after homologous (bovine sperm) and heterologous (murine sperm) EZPT using bovine ZPs co-incubated without OVGP1 (HoA, HeA) or with bovine OVGP1 (HoB, HeB).
Sperm was also previously co-incubated without OVGP1 or with bovine OVGP1 (HoAb*, HeAb*, HoBb*, HeBb*) or with murine OVGP1 (HeBm*).

| Group N | Sperm | ZPs OVGP1 | Sperm OVGP1 | TMC | Penetrated ZPs (%) | Sperm penetration inside ZP (mean) | Max. no. of sperm inside a single ZP | ZP with bound sperm (%) | Sperm bound to ZPs (mean) | Max. no. of sperm bound to a single ZP |
|---|---|---|---|---|---|---|---|---|---|---|
| HoA 15 | Bovine | - | - | 5.74±1.64 | 15 (100±0)[a] | 272 (18.13±1.32)[a] | 32±2.08 | 15 (100±0) | 338 (22.53±2.08)[a] | 25±2.51 |
| HoAb* 61 | Bovine | - | Bovine | 5.74±1.64 | 61 (100.0±10.00)[a] | 963 (15.80±1.18)[a] | 31±1.53 | 61 (100±0) | 1093 (17.97±3.01)[a] | 31±1.15 |
| HeA 60 | Murine | - | | 4.31±0.94 | 55 (91.67±8.15)[b] | 236 (4.24±0.52)[b] | 16±1.15 | 60 (100±0) | 338 (5.63±0.48)[b] | 21±1.00 |
| HeAb* 55 | Murine | - | Bovine | 4.31±0.94 | 49 (88.24±11.76)[b] | 198 (4.11±0.51)[b] | 14±1.00 | 55 (100±0) | 276 (5.00±0.31)[b] | 18±2.65 |
| HeAm* 58 | Murine | - | Murine | 4.31±0.94 | 52 (89.65±9.75)[b] | 233 (4.02±0.66)[b] | 15±1.73 | 58 (100±0) | 294 (5.08±0.32)[b] | 22±0.58 |
| HoB 14 | Bovine | Bovine | - | 5.74±1.64 | 14 (100±0)[b] | 232 (16.65±0.99)[a] | 33±1.52 | 14 (100±0) | 288 (20.88±3.88)[a] | 33±2.08 |
| HoBb* 59 | Bovine | Bovine | Bovine | 5.74±1.64 | 59 (100±0)[a] | 906 (15.34±0.66)[a] | 29±1.00 | 59 (100±0) | 1001 (16.87±2.34)[a] | 32±1.52 |
| HeB 16 | Murine | Bovine | - | 4.31±0.94 | 0 (0±0)[c] | 0 (0±0)[c] | 0±0 | 15 (94.44±9.62) | 78 (4.90±0.61)[b] | 8±0.57 |
| HeBb* 48 | Murine | Bovine | Bovine | 4.31±0.94 | 0 (0±0)[c] | 0 (0±0)[c] | 0±0 | 41 (84.81±10.67) | 115 (2,39±0.16)[c] | 11±1.52 |
| HeBm* 51 | Murine | Bovine | Murine | 4.31±0.94 | 2 (3.98±3.51)[c] | 2 (1±0)[c] | 1±0 | 51 (100±0) | 229 (4.48±0.57)[b] | 9±1 |

HoA = homologous EZPT with bovine sperm. HoAb*=homologous EZPT with bovine sperm (sperm was previously co-incubated with bovine OVGP1 for 30 min). HeA = heterologous EZPT with murine sperm. HeAb*=heterologous EZPT with murine sperm (sperm was previously co-incubated with bovine OVGP1 for 30 min). HeAm*=heterologous EZPT with murine sperm (sperm was previously co-incubated with murine OVGP1 for 30 min). HoB = homologous EZPT with bovine sperm (ZPs were co-incubated with bovine OVGP1 for 30 min). HoBb*=homologous EZPT with bovine sperm (ZPs were co-incubated with bovine OVGP1 for 30 min) (sperm was previously co-incubated with bovine OVGP1 for 30 min). HeB = heterologous EZPT with murine sperm (ZPs were co-incubated with bovine OVGP1 for 30 min). HeBb*=heterologous EZPT with murine sperm (ZPs were co-incubated with bovine OVGP1 for 30 min) (sperm was previously co-incubated with bovine OVGP1 for 30 min). HeBm*=heterologous EZPT with murine sperm (ZPs were co-incubated with bovine OVGP1 for 30 min) (sperm was previously co-incubated with murine OVGP1 for 30 min). TMC = total motile sperm count after capacitation calculated by multiplying the volume by concentration (million sperm/mL) and by the motility (%) after capacitation. Penetrated ZP refers to the mean number of ZPs penetrated by at least one sperm. ZPs with bound sperm was calculated as the mean number of ZP with, at least, one bound sperm. Sperm penetration was taken as the total number of sperm observed inside the zonas. Bound sperm was expressed as the mean number of sperm that remained bound to the ZP. 'N' refers to the total number of ZPs fertilized per treatment. Analysis was conducted by using a one-way ANOVA, followed by Tukey's post hoc test. Different superscripts (a, b, c) in the same column indicate significant differences (p<0.05; n=3). Values are expressed as the mean ± SD.

**Appendix 1—table 10.** Rates of sperm penetration and binding to bovine ZP after homologous (bovine sperm) and heterologous (human and murine sperm) EZPT using bovine ZPs co-incubated without OVGP1 (HoA, HeAh, HeAm) or with bovine OVGP1 (HoBb, HeBh, HeBm), human OVGP1 (HoC, HeCh, HeCm) or murine OVGP1 (HoD, HeDh, HeDm).

| Group N | Sperm | OVGP1 | TMC | Penetrated ZPs (%) | Sperm penetration inside ZP (mean) | Max. no. of sperm inside a single ZP | ZP with bound sperm (%) | Sperm bound to ZPs (mean) | Max. no. of sperm bound to a single ZP |
|---|---|---|---|---|---|---|---|---|---|
| HoA 67 | Bovine | - | 4.15±0.61 | 67 (100±0)[a] | 730 (10.89±0.40)[a] | 33±5.59 | 67 (100±0) | 918 (13.70±0.21)[a] | 25±2.51 |
| HeAh 30 | Human | - | 8,81±0.74 | 21 (70.0±10.00)[c] | 182 (8.66±0.68)[c] | 13±2.08 | 10 (33.33±5.77) | 18 (0.60±0.66)[b] | 3±0.58 |
| HeAm 83 | Murine | - | 8.76±1.07 | 80 (96.25±5.3)[a] | 300 (3.75±0.31)[c] | 21±2.12 | 83 (100±0) | 490 (5.90±0.81)[c] | 20±3.54 |
| HoB 70 | Bovine | Bovine | 4.15±0.61 | 69 (98.57±1.36)[a] | 708 (10.26±1.51)[a] | 29±4.30 | 70 (100±0) | 768 (10.97±1.31)[a] | 22±2.38 |
| HeBh 90 | Human | Bovine | 8,81±0.74 | 0 (0±0)[b] | 0 (0±0)[b] | 0±0 | 12 (13.33±3.33) | 12 (0.13±0.03)[b] | 1±0 |

*Appendix 1—table 10 Continued on next page*

*Appendix 1—table 10 Continued*

| Group N | Sperm | OVGP1 | TMC | Penetrated ZPs (%) | Sperm penetration inside ZP (mean) | Max. no. of sperm inside a single ZP | ZP with bound sperm (%) | Sperm bound to ZPs (mean) | Max. no. of sperm bound to a single ZP |
|---|---|---|---|---|---|---|---|---|---|
| HeBm 81 | Murine | Bovine | 8.76±1.07 | 3 (3.79±2.08)[b] | 3 (1±0)[b] | 1±0 | 36 (44.31±3.16) | 94 (1.17±0.28)[b] | 6±0.71 |
| HoC 84 | Bovine | Human | 5.03±0.66 | 49 (58.3±1.12)[c] | 235 (4.80±0.02)[c] | 11±1.41 | 84 (100±0) | 820 (4.80±0.02)[c] | 27±0.71 |
| HeCh 87 | Human | Human | 4.79±0.15 | 53 (61.11±7.85)[c] | 249 (4.73±0.73)[c] | 8±0.71 | 87 (100±0) | 832 (9.56±0.02)[c] | 27±1.41 |
| HeCm 30 | Murine | Human | 6.90±0.67 | 0 (0±0)[b] | 0 (0±0)[b] | 0±0 | 2 (6.67±0) | 2 (0.08±0.03)[b] | 1±0 |
| HoD 68 | Bovine | Murine | 4.35±0.46 | 43 (63.16±3.61)[c] | 199 (4.66±0.55)[c] | 7±0.71 | 68 (100±0) | 602 (8.86±0.20)[c] | 16±0.71 |
| HeDh 23 | Human | Murine | 4.79±0.15 | 0 (0±0)[b] | 0 (0±0)[b] | 0±0 | 1 (12.5±8.84) | 1 (0.06±0.09)[b] | 1±0.71 |
| HeDm 73 | Murine | Murine | 5.23±0.43 | 46 (63.23±7.55)[c] | 211 (4.59±0.13)[c] | 7±1.41 | 73 (100±0) | 980 (13.42±0.27)[a] | 25±3.53 |

HoA = homologous EZPT with bovine sperm. HeAh = heterologous EZPT with human sperm. HeAm = heterologous EZPT with murine sperm. HoB = homologous EZPT with bovine sperm (ZPs were co-incubated with bovine OVGP1 for 30 min). HeBh = heterologous EZPT with human sperm (ZPs were co-incubated with bovine OVGP1 for 30 min). HeBm = heterologous EZPT with murine sperm (ZPs were co-incubated with bovine OVGP1 for 30 min). HoC = homologous EZPT with bovine sperm (ZPs were co-incubated with human OVGP1 for 30 min). HeCh = heterologous EZPT with human sperm (ZPs were co-incubated with human OVGP1). HeCm = heterologous EZPT with murine sperm (ZPs were co-incubated with human OVGP1 for 30 min). HoD = homologous EZPT with bovine sperm (ZPs were co-incubated with murine OVGP1 for 30 min). HeDh = heterologous EZPT with human sperm (ZPs were co-incubated with murine OVGP1). HeDm = heterologous EZPT with murine sperm (ZPs were co-incubated with murine OVGP1 for 30 min). TMC = total motile sperm count after capacitation calculated by multiplying the volume by concentration (million sperm/mL) and by the motility (%) after capacitation. Penetrated ZP refers to the mean number of ZPs penetrated by at least one sperm. ZPs with bound sperm was calculated as the mean number of ZP with, at least, one bound sperm. Sperm penetration was taken as the total number of sperm observed inside the zonas. Bound sperm was expressed as the mean number of sperm that remained bound to the ZP. 'N' refers to the total number of ZPs fertilized per treatment. Analysis was conducted by using a one-way ANOVA, followed by Tukey's post hoc test. Different superscripts (a, b, c) in the same column indicate significant differences ($p<0.05$; n=3). Values are expressed as the mean ± SD.

**Appendix 1—table 11.** Rates of sperm penetration and binding to murine ZP after homologous (murine sperm) and heterologous (human and bovine sperm) EZPT using bovine ZPs co-incubated without OVGP1 (HoA, HeAh, HeAb) or with in vivo-matured murine OVGP1 (oocytes from superovulated females) (HoBb, HeBh, HeBb), murine OVGP1 (HoC, HeCh, HeCb), human OVGP1 (HoD, HeDh, HeDb) or bovine OVGP1 (HoE, HeEh, HeEb).

| Group N | Sperm | OVGP1 | TMC | Penetrated ZPs (%) | Sperm penetration inside ZP (mean) | Max. no. of sperm inside a single ZP | ZP with bound sperm (%) | Sperm bound to ZPs (mean) | Max. no. of sperm bound to a single ZP |
|---|---|---|---|---|---|---|---|---|---|
| HoA 56 | Murine | - | 8.76±1.08 | 56 (100±0)[b] | 848 (15.14±1.19)[b] | 29±5.50 | 56 (100±0) | 1042 (18.60±2.21) | 32±2.06 |
| HeAh 85 | Human | - | 6.12±0.17 | 60 (70.83±5.89)[c] | 177 (2.95±0.07)[c] | 9±0.71 | 79 (92.92±0.59) | 263 (3.32±0.31)[d] | 11±1.41 |
| HeAb 97 | Bovine | - | 3.67±0.19 | 56 (57.85±5.45)[c] | 167 (2.95±0.02)[c] | 6±0.00 | 91 (93.87±2.65) | 303 (3.32±0.40)[d] | 10±1.41 |
| HoB 19 | Murine | Murine | 5.35±0.07 | 19 (100±0)[b] | 228 (12.00±0.15)[b] | 31±7.78 | 19 (100±0) | 313 (16.47±0.64)[b] | 34±2.83 |
| HeBh 30 | Human | Murine | 6.12±0.17 | 1 (6.67±4.71)[a] | 1 (1±0)[a] | 1±0 | 3 (10.00±4.71) | 3 (1.00±0.00)[a] | 1±0 |
| HeBb 27 | Bovine | Murine | 4.26±0.09 | 1 (6.25±4.42)[a] | 1 (1±0)[a] | 1±0 | 3 (10.80±2.41) | 3 (1.00±0.00)[a] | 1±0 |
| HoC 99 | Murine | Human | 6.33±1.46 | 67 (67.49±2.39)[c] | 228 (3.40±0.24)[c] | 11±1.41 | 97 (98.00±2.82) | 449 (4.63±0.09)[d] | 19±0.71 |

*Appendix 1—table 11 Continued on next page*

*Appendix 1—table 11 Continued*

| Group N | Sperm | OVGP1 | TMC | Penetrated ZPs (%) | Sperm penetration inside ZP (mean) | Max. no. of sperm inside a single ZP | ZP with bound sperm (%) | Sperm bound to ZPs (mean) | Max. no. of sperm bound to a single ZP |
|---|---|---|---|---|---|---|---|---|---|
| HeCh 104 | Human | Human | 5.60±0.91 | 67 (64.39±2.32)[c] | 185 (2.76±0.02)[c] | 8±0.71 | 101 (10.38±0.54) | 1079 (10.68±0.54)[c] | 29±2.12 |
| HeCb 30 | Bovine | Human | 6.09±1.31 | 0 (0±0)[a] | 0 (0±0)[a] | 0±0 | 2 (6.67±0) | 2 (1.00±0)[a] | 1±0 |
| HoD 88 | Murine | Bovine | 5.29±0.43 | 56 (63.80±1.49)[c] | 219 (3.91±0.21)[c] | 8±0.71 | 87 (99.01±1.39) | 459 (5.28±0.62)[d] | 19±1.41 |
| HeDh 25 | Human | Bovine | 5.45±0.75 | 0 (0±0)[a] | 0 (0±0)[a] | 0±0 | 1 (3.33±4.71) | 1 (1.00±0.04)[a] | 1±0.71 |
| HeDb 90 | Bovine | Bovine | 4.35±0.46 | 54 (59.88±1.90)[c] | 182 (3.37±1.38)[c] | 7±0.71 | 90 (100±0) | 956 (10.62±0.08)[c] | 28±1.41 |

HoA = homologous EZPT with murine sperm. HeAh = heterologous EZPT with human sperm. HeAb = heterologous EZPT with bovine sperm. HoB = homologous EZPT with murine sperm (ZPs were co-incubated with murine OVGP1 for 30 min). HeBh = heterologous EZPT with human sperm (ZPs were co-incubated with murine OVGP1 for 30 min). HeBb = heterologous EZPT with bovine sperm (ZPs were co-incubated with murine OVGP1 for 30 min). HoC = homologous EZPT with murine sperm (ZPs were co-incubated with human OVGP1 for 30 min). HeCh = heterologous EZPT with human sperm (ZPs were co-incubated with human OVGP1). HeCb = heterologous EZPT with bovine sperm (ZPs were co-incubated with human OVGP1 for 30 min). HoD = homologous EZPT with murine sperm (ZPs were co-incubated with bovine OVGP1 for 30 min). HeDh = heterologous EZPT with human sperm (ZPs were co-incubated with bovine OVGP1). HeDb = heterologous EZPT with bovine sperm (ZPs were co incubated with bovine OVGP1 for 30 min). TMC = total motile sperm count after capacitation calculated by multiplying the volume by concentration (million sperm/mL) and by the motility (%) after capacitation. Penetrated ZP refers to the mean number of ZPs penetrated by at least one sperm. ZPs with bound sperm was calculated as the mean number of ZP with, at least, one bound sperm. Sperm penetration was taken as the total number of sperm observed inside the zonas. Bound sperm was expressed as the mean number of sperm that remained bound to the ZP. "N" refers to the total number of ZPs fertilized per treatment. Analysis was conducted by using a one-way ANOVA, followed by Tukey´s post hoc test. Different superscripts (a, b, c. d) in the same column indicate significant differences (p<0.05; n=3). Values are expressed as the mean ± SD.

**Appendix 1—table 12.** Rates of sperm penetration and binding to bovine ZP after homologous (bovine sperm) EZPT using bovine ZPs co-incubated with neuraminidase (NMase) and without NMase and with bovine OVGP1 or without bovine OVGP1.

| Group N | NMase | Bovine OVGP1 | TMC | Penetrated ZPs (%) | Sperm penetration inside ZP (mean) | Max. no. of sperm inside a single ZP | ZP with bound sperm (%) | Sperm bound to ZPs (mean) | Max. no. of sperm bound to a single ZP |
|---|---|---|---|---|---|---|---|---|---|
| HoA 55 | + | - | 5.89±0.22 | 2 (3.67±0.47)[b] | 2 (1±0.0)[b] | 1±0 | 53 (96.67±4.71) | 159 (2.89±0.35)[b] | 4±0.00 |
| HoAb 53 | + | + | 5.89±0.22 | 3 (5.89±3.45)[b] | 3 (1±0)[b] | 1±0 | 52 (98.27±2.44) | 202 (3.81±0.73)[b] | 8±0.71 |
| HoB 46 | - | - | 5.89±0.22 | 46 (100±0)[a] | 492 (10.62±0.82)[a] | 33±2.12 | 46 (100±0) | 759 (16.50±2.56)[a] | 33±2.82 |
| HoBb 45 | - | + | 5.89±0.22 | 45 (100±0)[a] | 499 (11.00±0.82)[a] | 32±1.41 | 45 (100±0) | 757 (16.82±2.34)[a] | 31±0.71 |

HoA = homologous EZPT with bovine sperm (ZPs were co-incubated with NMase at 10 U/mL in acetate buffer, PH 4.5 at 38 °C for 18 hr). HoAb = homologous EZPT with bovine sperm (ZPs were co-incubated with bovine OVGP1 for 30 min and then co-incubated with NMase diluted to 10μ/mL in acetate buffer, PH 4.5 at 38 °C for 18 hr). HoB = homologous EZPT with bovine sperm (ZPs were co-incubated with acetate buffer, PH 4.5 at 38 °C for 18 hr). HoBb = homologous EZPT with bovine sperm (ZPs were co-incubated with bovine OVGP1 for 30 min and then co-incubated with acetate buffer, PH 4.5 at 38 °C for 18 hr). TMC = total motile sperm count after capacitation calculated by multiplying the volume by concentration (million sperm/mL) and by the motility (%) after capacitation. Penetrated ZPs refers to the mean number of ZPs penetrated by at least one sperm. ZPs with bound sperm was calculated as the mean number of ZP with, at least, one bound sperm. Sperm penetration was taken as the total number of sperm observed inside the zonas. Bound sperm was expressed as the mean number of sperm that remained bound to the ZP. 'N' refers to the total number of ZPs fertilized per treatment. Analysis was conducted by using a one-way ANOVA, followed by Tukey´s post hoc test. Different superscripts (a, b) in the same column indicate significant differences (p<0.05; n=3). Values are expressed as the mean ± SD.

**Appendix 1—table 13.** Rates of sperm penetration and binding to murine ZP after homologous (murine sperm) EZPT using murine ZPs co-incubated with neuraminidase (NMase) and without NMase and with murine OVGP1 or without murine OVGP1.

| Group N | NMase | Murine OVGP1 | TMC | Penetrated ZPs (%) | Sperm penetration inside ZP (mean) | Max. no. of sperm inside a single ZP | ZP with bound sperm (%) | Sperm bound to ZPs (mean) | Max. no. of sperm bound to a single ZP |
|---|---|---|---|---|---|---|---|---|---|
| HoA 55 | + | - | 6.90±0.94 | 3 (5.45±2.16)[b] | 3 (1±0)[b] | 1±0 | 53 (96.15±5.43) | 257 (4.84±0.59)[b] | 9±0.71 |
| HoAm 53 | + | + | 6.90±0.94 | 3 (5.66±2.31)[b] | 3 (1±0)[b] | 1±0 | 50 (93.98±5.36) | 186 (3.50±1.40)[b] | 10±0 |
| HoB 28 | - | - | 6.90±0.94 | 28 (100±0)[a] | 345 (12.29±0.74)[a] | 33±1.99 | 28 (100±0) | 512 (18.28±0.67)[a] | 31±7.99 |
| HoBm 40 | - | + | 6.90±0.94 | 40 (100±0)[a] | 398 (9,97±0.8)[a] | 31±0,99 | 40 (100±0) | 363 (9.08±0.18)[a] | 30±0.98 |

HoA = homologous EZPT with murine sperm (ZPs were co-incubated with NMase diluted to 10µ/mL in acetate buffer, PH 4.5 at 38 °C for 18 hr). HoAm = homologous EZPT with murine sperm (ZPs were co-incubated with murine OVGP1 for 30 min and then co-incubated with NMase at 10 U/mL in acetate buffer, PH 4.5 at 38 °C for 18 hr). HoB = homologous EZPT with murine sperm (ZPs were co-incubated with acetate buffer, PH 4.5 at 38 °C for 18 hr). HoBm = homologous EZPT with murine sperm (ZPs were co-incubated with murine OVGP1 for 30 min and then co-incubated with acetate buffer, PH 4.5 at 38 °C for 18 hr). TMC = total motile sperm count after capacitation calculated by multiplying the volume by concentration (million sperm/mL) and by the motility (%) after capacitation. Penetrated ZPs refers to the mean number of ZPs penetrated by at least one sperm. ZPs with bound sperm was calculated as the mean number of ZP with, at least, one bound sperm. Sperm penetration was taken as the total number of sperm observed inside the zonas. Bound sperm was expressed as the mean number of sperm that remained bound to the ZP. 'N' refers to the total number of ZPs fertilized per treatment. Analysis was conducted by using a one-way ANOVA, followed by Tukey´s post hoc test. Different superscripts (a, b) in the same column indicate significant differences (p<0.05; n=3). Values are expressed as the mean ± SD.

**Appendix 1—table 14.** Rates of sperm penetration and binding to bovine ZP after homologous (bovine sperm) and heterologous (murine sperm) EZPT using bovine ZPs co-incubated with bovine OVGP1 previously treated with Neuraminidase (NMase) (HoA, HeA) and with non-treated bOVGP1 (HoB, HeB).
Control groups without ZPs treatment was added for both, homologous (HoC) and heterologous (HeC).

| Group N | Bovine OVGP1 | NMase treatment to bOVGP1 | TMC | Penetrated ZPs (%) | Sperm penetration inside ZP (mean) | Max. no. of sperm inside a single ZP | ZP with bound sperm (%) | Sperm bound to ZPs (mean) | Max. no. of sperm bound to a single ZP |
|---|---|---|---|---|---|---|---|---|---|
| HoA 15 | + | + | 5.25±0.60 | 76 (84.49±9.91)[a] | 473 (5.11±1.12)[a] | 22±1.53 | 87 (100±0) | 720 (8.02±1.06)[a] | 29±3.05 |
| HoB 87 | + | - | 5.25±0.60 | 15 (100±0)[b] | 254 (21.07±7.61)[b] | 29±3.05 | 15 (100±0) | 355 (23.67±7.11)[b] | 33±5.51 |
| HeA 82 | + | + | 5.34±0.05 | 45 (52.78±12.73)[c] | 123 (1.46±0.27)[c] | 6.00±1.00 | 65 (79.58±7.88) | 286 (3.51±0.31)[c] | 11±1.52 |
| HeB 15 | + | - | 5.34±0.05 | 0 (0±0)[d] | 0 (0±0)[d] | 0±0 | 4 (26.67±11.54) | 5 (0.33±0.23)[d] | 2±0.57 |
| HeC 20 | - | - | 5.34±0.05 | 18 (88.10±6.73)[a] | 113 (5.70±0.19)[a] | 13±2.12 | 20 (100±0) | 185 (9.98±2.61)[a] | 19±3.54 |

HoA = homologous EZPT with bovine sperm (ZPs were co-incubated with bovine OVGP1 for 30 min. Bovine OVGP1 was previously treated by incubation with NMase at 10U/mL in acetate buffer, PH 4.5 at 38°C for 18 hr). HoB = homologous EZPT with bovine sperm (ZPs were co-incubated with bovine OVGP1 for 30 min. Bovine OVGP1 was previously incubated in acetate buffer, PH 4.5 at 38°C for 18 hr). HeA = heterologous EZPT with murine sperm (ZPs were co-incubated with bovine OVGP1 for 30 min. Bovine OVGP1 was previously treated with NMase at 10U/mL in acetate buffer, PH 4.5 at 38°C for 18 hr). HeB = Heterologous EZPT with murine sperm (ZPs were co-incubated with bovine OVGP1 for 30 min. Bovine OVGP1 was previously treated in acetate buffer, PH 4.5 at 38°C for 18 hr). HeC = heterologous EZPT with murine sperm (ZPs were co-incubated with acetate buffer, PH 4.5 at 38°C for 18 hr). TMC = total motile sperm count after capacitation calculated by multiplying the volume by concentration (million sperm/mL) and by the motility (%) after capacitation. Penetrated ZPs refers to the mean number of ZPs penetrated by at least one sperm. ZPs with bound sperm was calculated as the mean number of ZP with, at least, one bound sperm. Sperm penetration was taken as the total number of sperm observed inside the zonas. Bound sperm was expressed as the mean number of sperm that remained bound to the ZP. 'N' refers to the total number of ZPs fertilized per treatment. Analysis was conducted by using a one-way ANOVA, followed by Tukey´s post hoc test. Different superscripts (a, b, c, d) in the same column indicate significant differences (p <0.05; n=3). Values are expressed as the mean ± SD.

