## [Editor Report · eLife Assessment]

This **valuable** study unravels the mechanisms underlying mammalian sperm-oocyte recognition and penetration, shedding light on cross-species interactions. It provides **solid** evidence that exposure of sperm to oviductal fluid or OVGP1 proteins from bovine, murine, or human sources imparts species-specific zona pellucida (ZP) recognition, ensuring that only sperm from the corresponding species can penetrate the ZP, regardless of its origin. These findings hold significant potential for reproductive biology, offering insights to enhance porcine in vitro fertilization (IVF), which frequently suffers from polyspermy, as well as advancing human IVF through improved intrinsic sperm selection.

---

## [Referee Report · Reviewer #1 (Public review)]

Summary:

This interesting manuscript first shows that human, murine, and feline sperm penetrate the zona pellucida (ZP) of bovine oocytes recovered directly from the ovary, although first cleavage rates are reduced. Similarly, bovine sperm can penetrate superovulated murine oocytes recovered directly from the ovary. However, bovine oocytes incubated with oviduct fluid (30 min) are generally impenetrable by human sperm.

Thereafter, the cytoplasm was aspirated from murine oocytes - obtained from the ovary or oviduct. Binding and penetration by bovine and human sperm was reduced in both groups relative to homologous (murine) sperm. However, heterologous (bovine and human) sperm penetration was further reduced in oviduct vs. ovary derived empty ZP. These data show that outer (ZP) not inner (cytoplasmic) oocyte alterations reduce heterologous sperm penetration as well as homologous sperm binding.

This was repeated using empty bovine ZP incubated, or not, with bovine oviduct fluid. Prior oviduct fluid exposure reduced non-homologous (human and murine) empty ZP penetration, polyspermy, and sperm binding. This demonstrates that species-specific oviduct fluid factors regulate ZP penetrability.

To test the hypothesis that OVGP1 is responsible, the authors obtained histidiine-tagged bovine and murine OVGP1 and DDK-tagged human OVGP1 proteins. Tagging was to enable purification following over-expression in BHK-21 or HEK293T cells. The authors confirm these recombinant OVGP1 proteins bound to both murine and bovine oocytes. Moreover, previous data using oviduct fluid was mirrored using bovine oocytes supplemented with homologous (bovine) recombinant OVGP1, or not. This confirms the hypothesis, at least in cattle.

Next, the authors exposed bovine and murine empty ZP to bovine, murine, and human recombinant OVGP1, in addition to bovine, murine, or human sperm. Interestingly, both species-specific ZP and OVGP1 seem to be required for optimal sperm binding and penetration.

Lastly, empty bovine and murine ZP were treated with neuraminidase, or not, with or without pre-treatment with homologous OVGP1. In each case, neuraminidase reduced sperm binding and penetration. This further demonstrates that both ZP and OVGP1 are required for optimal sperm binding and penetration.

In summary, the authors demonstrate that two mechanisms seem to underpin mammalian sperm recognition and penetration, the first being specific (ZP-mediated) and the second non-specific (OVGP1 mediated).

---

## [Referee Report · Reviewer #2 (Public review)]

Summary:

In the manuscript entitled "Oviductin sets the species-specificity of the mammalian zona pellucida", de la Fuente et al analyze the species specificity of sperm-egg recognition by looking at sperm binding and penetration of zonae pellucidae from different mammalian species and find a role for the oviductal protein OVGP1 in determining species specificity.

Strengths:

By combining sperm, oocytes, zona pellucida (ZP), and oviductal fluid from different mammalian species, they elucidate the essential role of OVGP1 in conferring species-specific fertilization.

Weaknesses:

Mice with OVGP1 deletion are viable and fertile. It would be quite interesting to investigate the species-specificity of sperm-ZP binding in this model. That would indicate whether OVGP1 is the only glycoprotein involved in determining species-specificity. Alternatively, the authors could immunodeplete OVGP1 from oviductal fluid and then ascertain whether this depleted fluid retains the ability to impede cross-species fertilization.

Comments on revisions:

This resubmission addresses most of my comments and concerns.

---

## [Referee Report · Reviewer #3 (Public review)]

Summary:

The authors submitted a second revised manuscript that reports findings from a series of experiments suggesting that bovine oviductal fluid and species-specific oviductal glycoprotein (OVGP1 or oviductin) from bovine, murine, or human sources modulate the species specificity of bovine and murine oocytes.

Strengths:

The study reported in the manuscript deals with an important topic of interest in reproductive biology.

Weaknesses:

The authors submitted a second revised manuscript. Some of the previous questions are considered inadequate. There are still several problematic issues that require the authors' attention.

---

## [Author Response]

The following is the authors’ response to the original reviews

**Reviewer #3 (Recommendations for the authors):**
Major concerns:P.6, lines 223-224: The sentence sounds like the authors produced all the OVGP1s by themselves in their laboratories, which is not completely true. The recombinant human and mouse OVGP1s were purchased from OriGene. It is suggested that the authors should state and explain clearly here which OVGP1 is produced by their laboratories and that recombinant human and mouse OVGP1s were obtained and purchased from Origene.

It is already clearly included in the M&M.

P6, lines 227-229: The authors stated that "Western blots of the three OVGP1recombinants indicated expected sizes based on those of the proteins: 75 kDa for human and murine OVGP1 and around 60 kDa for bovine OVGP1 (Fig. 4B and S6)." I pointed out in my last review report that the size of the recombinant human OVGP1shown by the authors in their manuscript is not in agreement with what has been published previously in literature regarding the molecular weight of native human OVGP1 as well as that of recombinant human OVGP1. The authors did not address the above concern adequately. In fact, recombinant human OVGP1 has been produced a few years ago (Reproduction (2016) 152:561-573) and it has been previously demonstrated that a single protein band of approximately 110-130 kDa was detected for both native human OVGP1 (see Microscopy Research and Technique (1995) 32:57-69) and recombinant human OVGP1 (Reproduction (2016) 152:561-573; Carbohydrate Research (2012) 358:47-55) using antibodies specific for human OVGP1. Molecular weight of the protein core or polypeptide of human OVGP1 is approximately 75 kDa, but the glycosylated form of native human OVGP1 and recombinant human OVGP1 is approximately 110-130 kDa. Therefore, the authors might have been using the recombinant core protein of human OVGP1 instead of the fully glycosylated recombinant OVGP1 in their study. The same concern also applies to the commercially obtained mouse recombinant OVGP1 used by the authors in their study. I would also like to mention that the mature and fully glycosylated OVGP1s in mammals vary in molecular weight (90-95 kDa in domestic animals; 110-150 kDa in primates; 160-350 kDa in rodents). Again, the 75kDa of mouse OVGP1 detected by the authors could be the core protein or polypeptide of mouse OVGP1 instead of the fully glycosylated mouse OVGP1.

In our study, as previously mentioned, we included commercially available recombinant proteins from Origene for human and murine OVGP1, which are produced in mammalian cells, and we also produced and purified bovine OVGP1 in mammalian cells. Therefore, these proteins should be properly glycosylated. Moreover, we performed Western blot assays favouring the blotting of higher molecular weight proteins, ensuring the optimal conditions for the assay. Additionally, we tested the size of OVGP1 from murine and bovine oviductal fluids on the same blot. During oestrus, the size of OVGP1 from oviductal fluids matches that of the recombinant proteins, and this band is downregulated during anoestrus, confirming the proper size of recombinant protein.

P.7, lines 236 and 237: Please provide a figure or source to support the statement "...as confirmed by proteomics of the bands along with PEAKS Studio v11.5 search engine peptide identification software."

It is included in the text the amount of unique peptides obtained by Proteomics for OVGP1 identification over all protein groups identified.

P.7, lines 243 to 245: The statement "...using rabbit polyclonal antibody to human OVGP1 for bOVGP1 and endogenous OVGP1, and mouse monoclonal antibody against Flag (DDK)-tag for hOVGP1 and mOVGP1." is confusing and might be inaccurate. First of all, I wondered why the authors did not use an antibody against bovine OVGP1 for the recombinant bOVGP1 instead of using a rabbit polyclonal antibody to human OVGP1. Secondly, what does the "endogenous OVGP1" refer to in the statement? Thirdly, the authors in their study used the commercially available recombinant human OVGP1 and recombinant mouse OVGP1 purchased from Origene. Based on the data sheet provided by Origene, the tag used for both recombinant human OVGP1 and recombinant mouse is C-Myc/DDK-tag and not Flag-tag. Can the authors explain these discrepancies?

Firstly, for the recombinant protein of bOVGP1 we used the same antibody that we used in the Western blot for all the proteins and oviductal fluids because we do not have anti-His tag working for Immunofluorescence (the one we had only worked for Western blot) and neither we do not have any antibody against bovine OVGP1. In the case of human and murine since we had anti-Flag antibody that worked for Western blot and for immunofluorescence, we used this one. However, as has been shown in our figure and supplementary material, the antibody against human OVGP1 works properly for both techniques (Western blot and Immunofluorescence). Secondly, endogenous OVGP1 is referred to the OVGP1 present in the oviductal fluid. Thirdly, as you can see in the datasheet of the protein, the recombinant proteins purchased from Origene contains a c-myc tag (EQKLISEEDL) some amino acids and a ddk-tag (DYKDDDDK). The sequence of ddk is the same of Flag-tag (DYKDDDDK). Since the proteins have both tags we used the antibody against Flag (or ddk) epitope.

P12, lines 429-432: The newly added statement at the end of the Discussion saying "Additionally, future studies would be valuable to investigate whether incubating oocytes with oviductal fluid (or OVGP1) could reduce polyspermy in porcine IVF and whether ZPs could be leveraged to naturally enhance sperm selection in human ICSI" is very concerning and requires further attention. The statement reflects that the authors do not keep pace with and do not pay attention to what has been published in literature regarding porcine and human OVGP1s. In fact, porcine oviduct-specific glycoprotein (OVGP1) has already been reported to reduce the incidence of polyspermy in pig oocytes (Biology of Reproduction (2000) 63:242-250). Porcine oviductal fluid, used in porcine IVF, has also been found to exert a beneficial effect on oocytes by reducing the incidence of polyspermy without decreasing the penetration rate. (Theriogenology (2016) 86:495-502). Therefore, the studies deemed valuable by the authors to be investigated in the future have, in fact, already been carried out two decades ago by several other laboratories. I am surprised the authors were not aware of these published work in literature. All the above should have been incorporated in the Discussion.

This sentence is modified in the discussion and the references are included.

Furthermore, as mentioned earlier, recombinant human OVGP1 has also been produced (Reproduction (2016) 152:561-573), and recombinant human OVGP1 has been found to increase tyrosine phosphorylation of sperm proteins, a biochemical hallmark of sperm capacitation, and potentiate the subsequent acrosome reaction (Reproduction (2016) 152:561-573) as well as increase sperm-zona binding (Journal of Assisted Reproduction and Genetics (2019) 36:1363-1377). These earlier findings should be incorporated into the Discussion.

Thank you for your comment, but in this work we had not performed any experimental setting related to tyrosine phosphorylation and despite is a very interesting topic is not directly related to this work.

P.19, lines 678-683: Since the human and mouse recombinant oviductin proteins were purchased from Origene, the authors should be aware of the fact that these commercially available recombinant OVGP1s might not be fully glycosylated. While I appreciate the fact that the authors wanted to briefly describe how the human and mouse recombinant OVGP1s were prepared by the manufacturer, I strongly suggest that the authors should contact Origene, the manufacturer, for all information regarding the procedures for producing the human and mouse recombinant oviductin proteins. For example, the authors stated on lines 680-681 that "A sequence expressing FLAG-tagged epitope proteins (DYKDDDDK) was cloned into an expression vector." According to the data sheet provided by Origene, it appears that both human and recombinant oviductin proteins are C-Myc/DDK-tagged and not FLAG-tagged.

Thank you for your comment, as according to the sequence of Flag-tag it is matching with the sequence of the tag in the datasheet corresponding to DDK (this is in detail in previous comment). Besides, the protein is tagged also by C-Myc tag. Among both tags, the antibody selected to detect it was anti-Flag tag.

P.19, lines 692-697: The description of the primary and secondary antibodies used for detection of the various recombinant OVGP1s is also very confusing and not clearly presented. For example, it is mentioned here that "...membranes were...incubated with anti-OVGP1 rabbit monoclonal antibody for OVGP1,..". What specifically does "OVGP1" refer to here? The authors then stated that anti-Histamine Tag antibody was used to detect bOVGP1 and mOVGP1 and anti-Flag antibody was used to detect hOVGP1. As pointed out earlier, the human and mouse recombinant OVGP1s were produced using C-Myc/DDK tag and not His-tag or Flag-tag. Can the authors clarify these discrepancies?

We apologise for the complexity of the antibodies, we included in this paragraph the ones used to Western blot for both figures: anti- human OVGP1 was used for the principal figure that contains the three recombinant proteins and oviductal fluids; and the anti-Histidine and anti-Flag antibodies that are included in supplementary figure, specifically for recombinant bovine OVGP1 (Histidine tag) and for recombinant murine and human OVGP (DDK tag). A clarifying sentence has been included in the text.

P.31, lines 1143-1149: Figure 10 is not mentioned anywhere in the main text of the manuscript. Rewrite the second half of the sentence "...; being this specificity lost when OVGP1 is heterologous to the ZP (right diagram)." Which sounds awkward and grammatically not correct.

The figure is already mentioned in the text, thank you for your comment. The sentence is also corrected.

Other comments: P.1, the statement of "All authors contributed equally to this work" on line 14 can be deleted because detailed and specific contributions from each authors are listed in lines 1009-1017 on page 27.

Both authors contributed equally to this work, now is clear in authors contribution section.

P.2, lines 43 and 44: Do the authors mean "sperm-oocyte binding protein" instead of "sperm-oocyte fusion protein" in the sentence? "Fusion protein" is a protein composed of two or more domains encoded by different genes, or a hybrid molecule created by combining two different proteins for various purposes. I believe the term "fusion protein" is wrongly used in the sentence which should be rephrased with a proper term.

Done.

P2, line 73: Remove the comma after the word "Both".

Done.

P.5, line 179: "...mice ZP..." should be written as "...mouse ZP...".

Done.

P.6, heading of 3rd paragraph on line 207: The term "binding" will be a better term than "fusion" used in the heading because the results do not actually show the fusion of the OVGP1 proteins with the ZP glycoprotein. Instead, binding of the OVGP1 proteins to the ZP occurred.

Done.

P.6, lines 215-217: Authors, please provide a reference or references to support the statement "Region A, corresponding to the amino acid end, shows high identity among monotremes, marsupials and placentals."

In the text was indicated a review (29) which includes the supporting idea of this statement for Figure 4. Moreover, we have included some if the references used for the description of the domains when performing the sequence alignment of Figure S5.

P.6, line 230 and line 233 on P.7: Authors, please be consistent in the use of either American English or British English. The word "oestrus" is British English whereas "estrus" is American English.

Done.

P.7, line 264: The word "sticking" used here means non-specific binding. I believe the author means specific binding here. If so, a more appropriate word should be used here instead of "sticking".

Done.

P.7, lines 267-269: This newly added sentence sounds very awkward and should be completely rewritten.

Done.

P.8, line 288: This reviewer finds it difficult to understand the meaning of the heading. The heading should be rephrased to bring out exactly what the authors want to say in well-written English.

Done.

P.8, line 290: The word "would" should be replaced by "could" in the sentence.

Done.

P.13, line 437: Authors, please provide the location of Sigma-Aldrich.

Done.

P.13, line 457: Here, the authors used "1800 rpm" to indicate the centrifugation speed but used the g-force elsewhere in the Materials and Methods. Please be consistent. The g-force is preferred.

Done.

P.14, lines 483-485: The procedure of sacrificing the cats should be provided in the Materials and Methods

Cats weren’t sacrificed they were vasectomized. It is now included in the text.

P.17, line 628: "...the ZPs were exposed or no exposed to..." should be written as "...the ZPs were either exposed or not exposed to...".

Done.

P.17, line 629: "...each groups were incubated with..." should be "...each group was incubated with...".

Done.

P.19, line 700: "As loading control, was used the primary antibody....." is not a complete sentence and it needs to be rewritten.

Done.

P.20, lines 744-754: For scanning electron microscopy and image processing, the procedures of prior treatment of the oocytes with and without oviductal fluid and OVGP1 should be included here.

Done.

P.21, line 756: It is stated here that "Two hundred isolated ZPs were treated with Clostridium perfringens neuraminidase....". However, it is not clear whether two hundred isolated ZPs of both porcine and murine ZPs were treated. Authors, please clarify.

We used 200 isolated ZPs of each specie, bovine and murine. It is classified in the text.

P.28, lines 1039 and 1040: The author only mentioned the use of bovine and murine sperm here. What about human sperm?

Done.

P.29, line 1076: "...in mammalian cells..." is very vague. Be specific what exactly the mammalian cells were.

Done.

P.29, line 1079: "Oviductal fluid from ovulated cows or anoestrus cows." is not a complete sentence and it needs to be rewritten.

Done.